# HIV-1-mediated insertional activation of *STAT5B* and *BACH2* trigger viral reservoir in T regulatory cells

Daniela Cesana[1], Francesca R. Santoni de Sio[1], Laura Rudilosso[1], Pierangela Gallina[1], Andrea Calabria[1], Stefano Beretta[2,3], Ivan Merelli[3], Elena Bruzzesi[4], Laura Passerini[1], Silvia Nozza[4], Elisa Vicenzi[5], Guido Poli[6,7], Silvia Gregori[1], Giuseppe Tambussi[4] & Eugenio Montini[1]

HIV-1 insertions targeting *BACH2* or *MLK2* are enriched and persist for decades in hematopoietic cells from patients under combination antiretroviral therapy. However, it is unclear how these insertions provide such selective advantage to infected cell clones. Here, we show that in 30/87 (34%) patients under combination antiretroviral therapy, *BACH2*, and *STAT5B* are activated by insertions triggering the formation of mRNAs that contain viral sequences fused by splicing to their first protein-coding exon. These chimeric mRNAs, predicted to express full-length proteins, are enriched in T regulatory and T central memory cells, but not in other T lymphocyte subsets or monocytes. Overexpression of BACH2 or STAT5B in primary T regulatory cells increases their proliferation and survival without compromising their function. Hence, we provide evidence that HIV-1-mediated insertional activation of *BACH2* and *STAT5B* favor the persistence of a viral reservoir in T regulatory cells in patients under combination antiretroviral therapy.

[1] San Raffaele Telethon Institute for Gene Therapy (SR-Tiget), IRCCS, San Raffaele Scientific Institute, Milan 20132, Italy. [2] Department of Informatics, Systems and Communication, University of Milano—Bicocca, Viale Sarca 336, Milan 20126, Italy. [3] National Research Council, Institute for Biomedical Technologies, Via Fratelli Cervi 93, Segrate 20090, Italy. [4] Department of Infectious Diseases, IRCCS, San Raffaele Scientific Institute, Milan 20132, Italy. [5] Viral Pathogens and Biosafety Unit, Division of Immunology, Transplantation and Infectious Diseases, IRCCS, San Raffaele Scientific Institute, Milan 20132, Italy. [6] AIDS Immunopathogenesis Unit, Division of Immunology, Transplantation and Infectious Diseases, IRCCS, San Raffaele Scientific Institute, Milan 20132, Italy. [7] Vita-Salute San Raffaele University School of Medicine, Milan 20132, Italy. Francesca R. Santoni de Sio and Laura Rudilosso contributed equally to this work. Correspondence and requests for materials should be addressed to D.C. (email: cesana.daniela@hsr.it) or to E.M. (email: montini.eugenio@hsr.it)

Viruses have evolved strategies to exploit the host machinery for their own propagation at every step of their replication cycle. It has been demonstrated in animals that several retroviruses take advantage of insertional mutagenesis to activate proto-oncogenes, leading to cell transformation and ultimately cancer, thus favoring viral spread and persistence in the host[1].

For lentiviruses such as HIV-1, expe evidences and novel observations suggest that its integration in the host genome could actually result in insertional mutagenesis. In hematopoietic cells, HIV-1 and lentiviral vectors (LVs) preferentially integrate within the transcription unit of expressed genes[2] and may induce aberrant RNA splicing mechanisms leading to the formation of chimeric transcripts harboring HIV sequences fused to cellular exon sequences[3–5]. Moreover, LVs with active long terminal repeats (LTRs) are able to effectively activate cancer-related genes through promoter insertion and thus inducing neoplastic transformation[6, 7]. Finally, a significant enrichment of proviral integrations targeting some cancer-related genes, such as *BACH2* and *MKL2* and others, has been observed in peripheral blood mononuclear cells (PBMC) and CD4$^+$ T lymphocytes isolated from HIV-infected individuals under combination antiretroviral therapy (cART)[8–10]. These data suggest that HIV-1, similarly to onco-retroviruses, could exploit insertional mutagenesis to activate or inactivate cancer-related genes, leading to the clonal expansion of the infected cell and thus favoring its persistence in the host. However, it is currently unknown how proviral integrations may cause the deregulation of these cellular genes and if the physiological consequences of this deregulation may result in oncogenesis or another phenotype that is selected in these specific conditions.

By retrieving HIV-1 insertion sites in a European cohort of HIV-1-infected patients, we found that *BACH2* and *STAT5B* were the two most frequently targeted genes. Since most of the viral insertions within the transcriptional unit of *BACH2* and *STAT5B* clustered in a small genomic window and were in the same orientation of the targeted gene transcription, we hypothesized that the HIV-1 LTR could directly control the expression of these genes by a mechanism known as promoter insertion and drive the formation of chimeric mRNA transcripts containing viral HIV-1 sequences fused by splicing to the first protein-coding exon of the targeted gene. By performing RT-PCR on the mRNA obtained from PBMCs of a large cohort of HIV patients ($N = 87$), we found in 34% of patients the predicted chimeric transcripts, putatively encoding for unaltered full-length BACH2 or STAT5B proteins. Interestingly, these transcripts were specifically enriched in T central memory and T regulatory (Treg) cells, two cellular compartments that have been proposed to be the main reservoir of HIV virus. Finally, we demonstrated that forced expression of these two transcription factors in Treg cells, purified from healthy donors (HDs), increased their proliferation rate without impacting on their immune-suppressive function. In conclusion, our data indicate that in a large portion of HIV patients under cART, HIV/*BACH2*, or *STAT5B* chimeric transcripts are expressed in Treg cells and such expression could favor the persistence of this important HIV cellular reservoir.

## Results

***BACH2* and *STAT5B* are highly targeted genes in HIV-1 patients**. In order to investigate the biological role of HIV-1-mediated insertional mutagenesis, we first attempted to characterize the HIV-1 integration profile in PBMC from a cohort of 54 HIV-1-infected individuals under cART followed in our Institute and described in Tambussi et al[11]. In this cohort 50% of patients under cART treatment had low levels of viremia

and in 29 patients (54%) the cART treatment was supplemented by IL-2 administration for 12 months[11] (Supplementary Table 1).

For integration site retrieval, linear amplification-mediated (LAM)-PCR was used to retrieve the viral/cellular genome junctions that were sequenced using the Illumina platform. Sequences were then mapped by a dedicated bioinformatics pipeline[12], previously used for the study of two LV-based gene therapy clinical trials[13, 14]. By this approach, a total of 13,671 HIV-1/cellular genomic junctions were retrieved, corresponding to 198 HIV-1 integration sites univocally mapped on the human genome (Supplementary Table 1). The genomic distribution of integration sites in our data set followed the known tendency of HIV-1 and replication-defective LVs to integrate within gene bodies and gene dense regions[13, 14] (Supplementary Fig. 1a, b). The genomic distribution in our integration data set was then compared to three previously published data sets from HIV-1-infected patients under cART[8–10] and to five data sets of non-replicating HIV-1-derived LVs used to transduce ex vivo CD4$^+$ T cells and hematopoietic stem cells (HSC) for gene therapy clinical trials[13–17]. With the exception of the HIV-1 data sets from Maldarelli et al.[9] and Wagner et al.[10], the six remaining data sets were reanalyzed with our bioinformatics pipeline[12] to harmonize the integration mapping data. The genomic distribution of integration sites of the different data sets was compared using intervals of 1 Mb by the Genomic Hyperbrowser analysis tool (v2.0b). No significant differences in the global genomic distribution of integration sites were found compared to Maldarelli et al., Wagner et al., data sets obtained from fully suppressed patients (Supplementary Table 2). In the data set from Ikeda et al., a region on chromosome 6 spanning from nt 900,000 to 100,000,000, containing *BACH2*, was significantly over targeted compared to our data set (false discovery rate, FDR 10% adjusted $p$-value $= 1.6 \times 10^{-5}$).

In our study, out of the 198 integrations retrieved, 3 targeted *BACH2* and 4 targeted *STAT5B* (distributed in seven patients), a targeting frequency significantly higher than expected, as determined by Montecarlo simulations[18]. Moreover, most of the insertions targeting these genes were in the same orientation of gene transcription and landed upstream of the first protein-coding exon (Supplementary Fig. 1c, d). Although the four insertions targeting *STAT5B* were retrieved only from patients that received the IL-2 treatment, its targeting frequency was not significantly different when compared to the control group without IL-2 treatment. Moreover, *BACH2* and *STAT5B* were among the top three targeted genes in all the HIV-derived integration data sets studied (Supplementary Table 3), indicating that the level of viral replication and the cART regimen adopted did not significantly impact on the overall distribution of HIV insertion sites. The targeting frequency of *BACH2* and *STAT5B* in our and the HIV-derived data sets was always significantly higher than in the LV data sets from ex vivo transduced CD4$^+$ T cells and HSC gene therapy clinical trials ($p < 0.0001$ by Fisher exact test, Supplementary Table 1, Supplementary Fig. 1e, f). Only, the Ikeda et al. data set showed a significantly higher targeting frequency for *BACH2* among the four data sets from HIV-infected patients under cART (Supplementary Table 4, Supplementary Fig. 1e, f), in agreement with the results obtained with the comparative analysis on the integration distribution at the genomic-wide level by the Genomic Hyperbrowser analysis tool (Supplementary Table 2). Moreover, in all the HIV data sets, viral insertions in *BACH2* were always found in the same transcriptional orientation as the host gene, while they were randomly distributed in the LV-derived data sets (Supplementary Table 4). Differently, in our and Ikeda's data sets almost all insertions targeting *STAT5B* were in the same orientation of gene transcription, while in the Maldarelli et al. and Wagner et al. data

sets this bias was not observed (Supplementary Table 4). Overall, these data indicate that *BACH2* and STAT5B insertions are enriched by a mechanism of insertional mutagenesis.

**Identification of chimeric mRNAs in HIV-1 patients**. The specific clustering of proviral integrations upstream the first protein-coding exon and the coherent orientation of the viral integrations in the same direction of *BACH2* and *STAT5B* gene transcription resemble a pattern found in insertional mutagenesis screenings performed with retroviruses, transposon, and geno-toxic LVs, in which the viral promoter drives the expression of oncogenes by read-through transcription and formation of aberrant splicing events[1, 7, 19, 20]. Therefore, we postulated that HIV-1 insertions targeting *BACH2* or *STAT5B* could similarly drive the expression of chimeric transcripts containing HIV-1 sequences fused by splicing to *STAT5B* or *BACH2* exons downstream the integration site. To address this hypothesis, we designed two oligonucleotide primers complementary to the HIV-1 LTR sequences paired to two oligonucleotides complementary to the coding exons 6 and 7 of *BACH2* or to two different portions of the exon 4 of *STAT5B*. This nested RT-PCR strategy should allow to specifically amplify HIV/*BACH2* or HIV/*STAT5B* chimeric transcripts that would be generated by HIV-1 insertions mapping in different positions of the targeted gene intron/s (Fig. 1a, b and Supplementary Methods). We thus applied this RT-PCR strategy on total RNA from PBMCs of three patients in which we previously found HIV insertions in *STAT5B* (JID.63 and JID.29, Supplementary Fig. 1) and *BACH2* (JID.87, Supplementary Fig. 1). In all patients, an RT-PCR product of the expected size (500 bp) was obtained (Fig. 1). Sequence analysis of the RT-PCR products showed that the transcripts contained the 5′ HIV-1 LTR sequences extended until the canonical viral splice donor (SD) site and then correctly fused by splicing to the first protein-coding exon of *STAT5B* (exon 2) and *BACH2* (exon 6) (Supplementary Fig. 2). Moreover, a HIV/*STAT5B* chimeric transcript was also amplified from a patient (JID.87) that did not show insertions targeting *STAT5B* in the previous integration site

analysis, suggesting that the RT-PCR for this specific transcripts could better suited to reveal insertional mutagenesis events that are instead missed by integration site retrieval approach. We therefore expanded the RT-PCR for analysis to detect HIV/*STAT5B* and HIV/*BACH2* chimeric transcripts on a second cohort of 87 patients, for which integration data was not available. This cohort was composed by 48 treatment-naïve patients (NT) that received only Non-Nucleoside Reverse Transcriptase Inhibitors cART regimen and 39 multi-experienced cART-treated patients (EXP) who received both Nucleoside and Non-Nucleoside Reverse Transcriptase Inhibitors cART regimen (Supplementary Table 5, and Methods). From this analysis we found that 27 patients were positive for the presence of HIV/*STAT5B* transcripts (31%, Fig. 1c, $N = 12$ in NT and $N = 15$ in EXP), 8 patients were positive for HIV/*BACH2* transcripts (9.2%, Fig. 1d, $N = 4$ in NT and $N = 4$ in EXP), and 5 patients (5.7%) expressed both (Fig. 1 and Supplementary Fig. 2, Supplementary Table 5). No biases for the presence of these chimeric transcripts were found in treatment-naïve vs. multi-experienced patients by Fisher exact test comparison. Sequence analysis of the RT-PCR products amplified from this cohort of patients confirmed that chimeric transcript for both genes were generated by a mechanism of promoter insertion (Supplementary Fig. 2).

**Enrichment of HIV/*STAT5B* and HIV/*BACH2* mRNAs in Treg cells**. We then investigated whether these chimeric transcripts could be preferentially expressed in different T lymphocyte subsets and/or monocytes. Therefore, nine patients who tested positive for HIV/*STAT5B* and/or HIV/*BACH2* chimeric transcripts in the RT-PCR screening on PBMCs were selected for further molecular analyses. Specifically, RT-PCR was used to test the presence of HIV/*STAT5B* transcripts in Treg cells, T effector (Teff), T naïve, T stem cell memory (Tscm), T central memory (Tcm), T effector memory (Tem), total CD8+ T lymphocytes and monocytes purified by fluorescence activated cellsorting (FACS) sorting from the blood of two patients (ID_77 and 96) and in

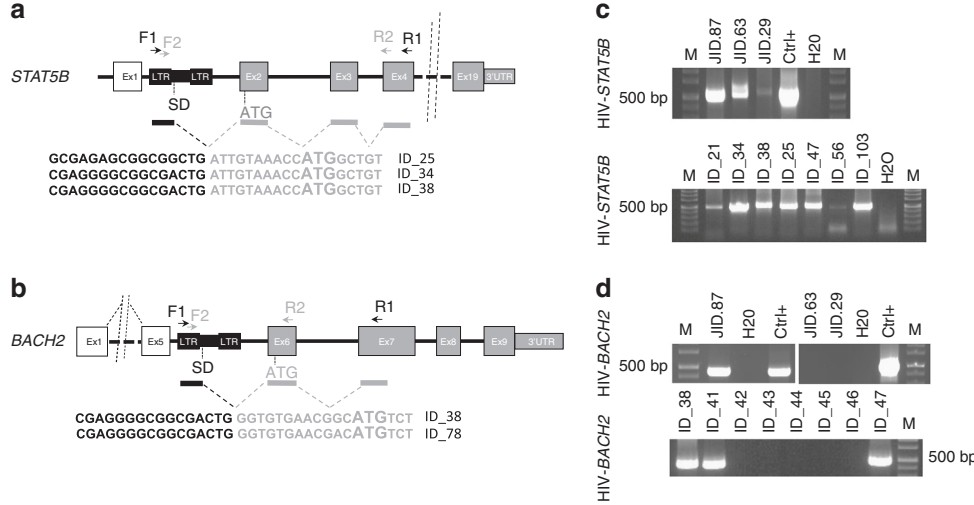

**Fig. 1** Identification of the chimeric HIV/*BACH2* and HIV/*STAT5B* transcripts in patients under cART. **a**, **b** Scheme of the RT-PCR strategy devised to amplify putative chimeric HIV/*STAT5B* (**a**) or HIV/*BACH2* (**b**) transcripts; *white* and *gray boxes* represent non-coding and coding exons (Ex), respectively. The exon numbers and the position of the first ATG codon are indicated. The integrated provirus, the LTR, and the 5′ major viral SD site are depicted in *black*. The *arrows* represent the position of the primer pairs used for RT-PCR. The *black* and *gray bars* indicate the amplified cDNA sequence from the provirus and host genome, respectively. The *dashed lines* indicate the splicing events. At the bottom of each panel, representative examples of the sequences of the RT-PCR products are shown. **c**, **d** Report agarose gel electrophoresis of RT-PCR products of either HIV/*STAT5B* or HIV/*BACH2* obtained from PBMC of HIV-1-infected patients using the primers indicated in **a**, **b**; the amplicon size is ~500 bp (M molecular size markers). For each sample, RT-PCR for *GAPDH* was also performed as positive control of amplification (data not shown)

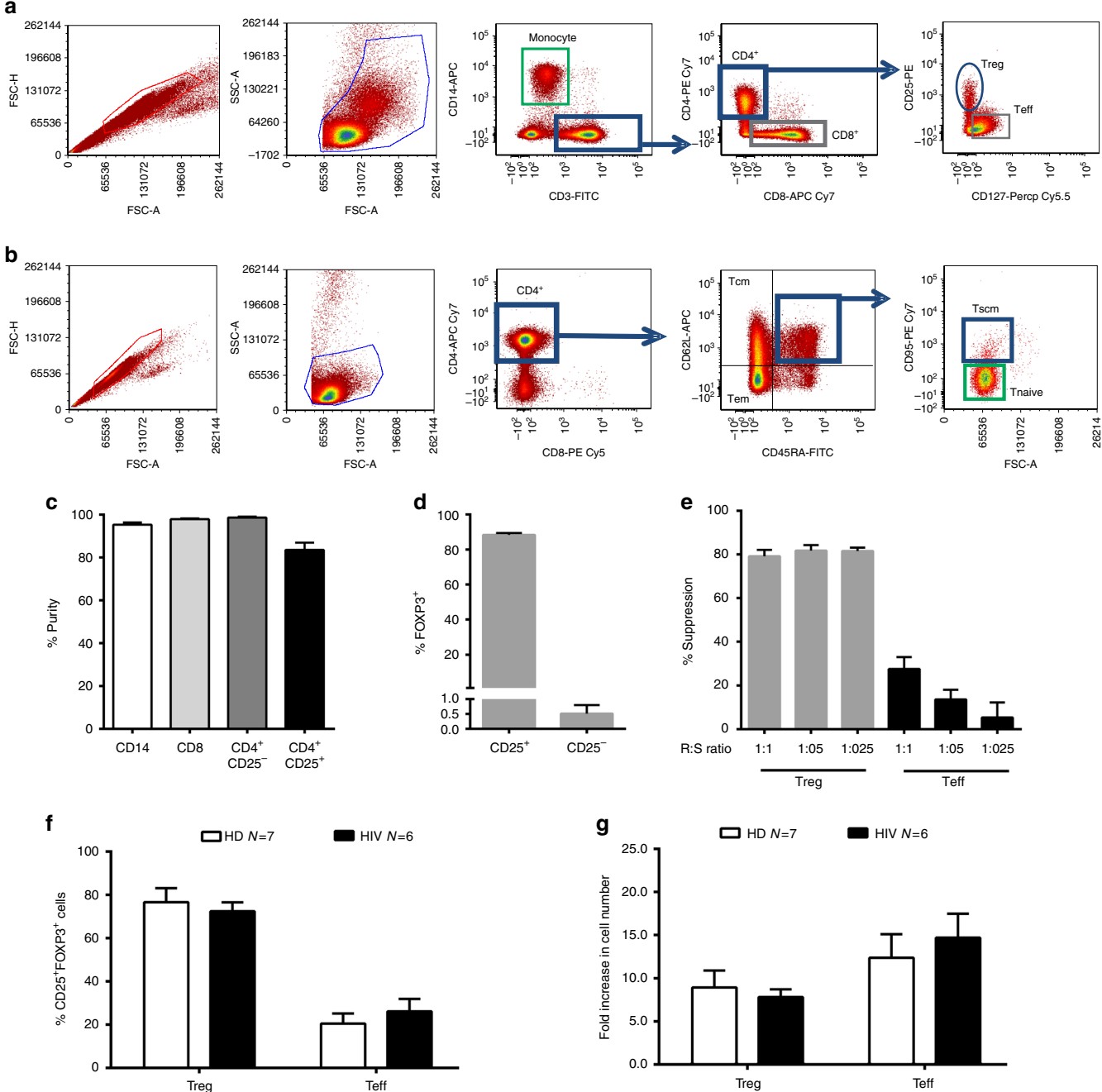

**Fig. 2** Isolation of different T lymphocyte subsets and monocytes from HIV-1-infected patients under cART. **a** Representative FACS dot plots illustrating the sequential gating process used for the identification and cell sorting strategy of Treg (CD4$^+$CD25$^+$CD127 low), Teff (CD3$^+$CD4$^+$CD25$^-$), monocytes (CD14$^+$), and CD8$^+$ cells; **b** FACS dot plots showing the sequential gating process used for the identification and cell sorting isolation of memory T cell subsets. Each memory cell subsets was identified by the following markers: CD4$^+$CD45RA$^+$CD62L$^+$CD95$^+$ for T stem cell memory (Tscm); CD4$^+$CD45RA$^+$CD62L$^+$CD95$^-$ for T-naïve; CD4$^+$CD45RA$^-$CD62L$^+$ for T central memory (Tcm); and CD4$^+$CD45RA$^-$CD62L$^-$ for T effector memory (Tem); **c** Assessment of the purity of the isolated monocyte (CD14$^+$), CD8$^+$ cells, CD4$^+$ and CD4$^+$CD25$^+$ cells purified by magnetic cell isolation, $N = 6$; **d** Percentage of FOXP3$^+$ cells measured in CD25$^+$ and CD25$^-$ fraction after FACS sorting or magnetic cell isolation, $N = 10$; **e** Inhibition of proliferation (analyzed by flow cytometry for e-Fluor dilution and indicated as % Suppression vs. Responder alone) of anti-CD3/CD28 activated allogeneic Responder cells (R, CD4$^+$ CD25$^-$) and cultured in presence of different doses (Ratio R:S were 1:1, 1:05, and 1:0.25) of purified and expanded Treg and Teff cells (S), $N = 10$. **f** Relative percentage of FOXP3$^+$ cells measured in Treg and Teff cells purified from the blood of HIV-infected patients or from HDs and expanded in culture for 10 days; **g** Fold increase in the amount Treg or Teff cells purified from the blood of HIV-infected patients or HD and expanded in culture for 10 days, $N$ is indicated. In each panel, data are represented as Mean ± SEM

Treg, Teff, CD8$^+$ T cells and monocytes purified by magnetic beads isolation from four patients (ID_25, 56, 63 and 76) (Fig. 2 and Supplementary Table 3). Moreover, the levels of the HIV/ *STAT5B* and HIV/*BACH2* chimeric transcripts were quantitatively measured by droplet digital (dd)-PCR on in vitro expanded

Treg and Teff cells from six patients (ID_25, 77, 76, 32, 38 and 71) (Fig. 2). HIV/*STAT5B* transcripts were found in the purified Treg cells of six out of six patients and in Teff cells of two out six donors (Fig. 3a, b). Moreover, in two out of two patients tested, we detected HIV/*STAT5B* transcripts in Tcm cells (Fig. 3a),

whereas Tscm, total CD8[+] T lymphocytes and monocytes were negative. The detection of the transcript in the Tcm compartment could be due to the presence of CD25[+] T cells (that include Treg cells) (Supplementary Fig. 3), or to the fact that Treg cells might differentiate from memory T cells in the periphery[21, 22] or both.

dd-PCR on the RNA isolated from Treg and Teff cells confirmed the enrichment of the HIV/*STAT5B* and HIV/*BACH2* chimeric transcripts in Treg cells on all patients (six out of six for HIV/*STAT5B* and three out of three for HIV/*BACH2*) (Fig. 3c, d and Supplementary Fig. 4). This quantitative analysis showed that wild-type *STAT5B* and *BACH2* transcripts were expressed at comparable level in Treg and Teff cells, while the chimeric transcripts were expressed at a significantly higher level in Treg cells (Fig. 3e, f and Supplementary Fig. 4), indicating that the different expression levels are not due to a different transcriptional regulation of the host gene between these two cell subsets. Furthermore, in all patients tested the chimeric transcripts were found in PBMCs and in Treg cells that were collected from the same individuals >6 years apart, indicating that cells expressing these chimeric transcripts persist over the years. We did not find any differences in the overall proportions of CD4[+] Treg cells in the HIV patients expressing at least one of the chimeric transcripts compared to normal donors (Supplementary Fig. 5).

**Chimeric mRNAs are generated by multiple viral haplotypes.** Since during infection the HIV-1 genome acquire mutations at relatively high frequency and generates a highly diverse viral population in patients, we reasoned that we could take advantage of the high genetic diversity of the integrated HIV-1 genome to get insights on the number of different cellular clones expressing HIV/*STAT5B* chimeric transcripts. In our rationale the presence of sequence variants (haplotypes) in the HIV-1 portion of the chimeric transcripts will indicate that these were generated by different viral haplotypes in independent infection events occurring in different cell clones. We thus performed Illumina sequencing of the amplified RT-PCR products of the chimeric transcripts from total PBMC of 10 patients, and from different T cell subsets of 6 patients (analyzed previously by RT-PCR, ID_25, 56, 63, 77, 76 and 96). Overall, we obtained an average of 5816 sequencing reads (ranging from 620 to 37010 reads) for each sample (Supplementary Table 6). The HIV-1 portion was then aligned to the reference HIV-1 sequence (HXB2 from the HIV database, www.hiv.lanl.gov) spanning from the U5 to the major SD signal (212 bp), while the *STAT5B* portion was aligned to the reference sequence spanning the exons 2, 3 and 4 (295 bp). The Propagating Dirichlet Process Mixture Model was then used to infer the number and frequency of the different HIV-1 haplotypes contained in each library[23]. While, in the *STAT5B* portion only a

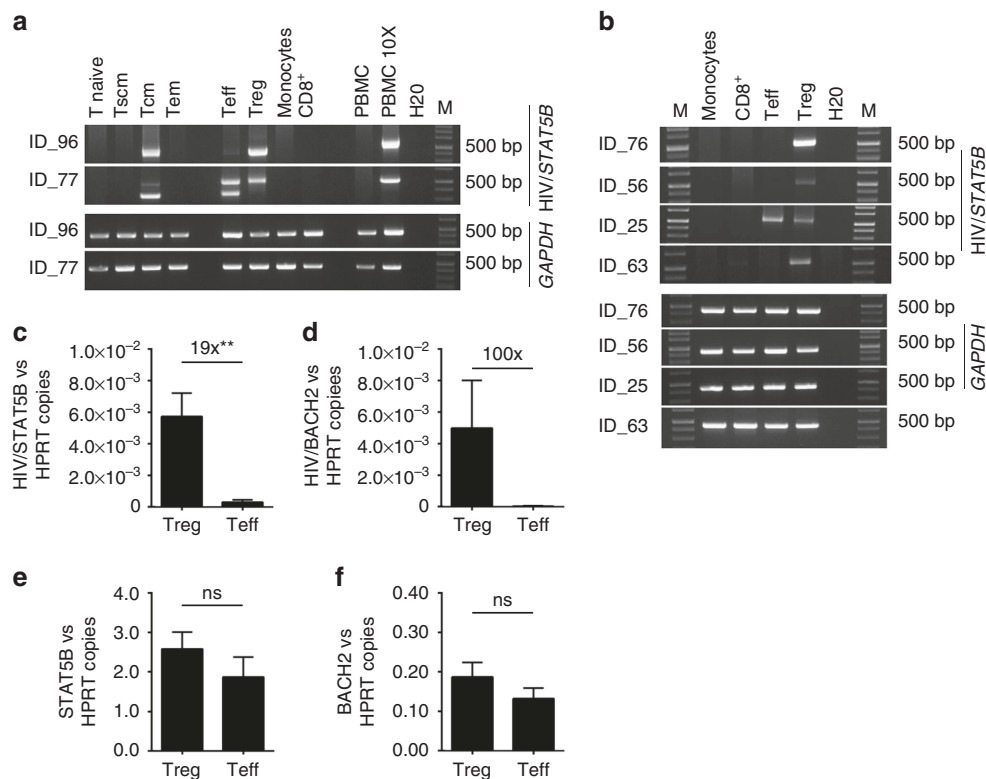

**Fig. 3** Tracking and quantification of HIV/*STAT5B* and HIV/*BACH2* chimeric transcripts in different hematopoietic cell subsets obtained from HIV patients under cART. **a** *Top panel*, agarose gel electrophoresis of the RT-PCR products (using the primers indicated in Fig. 1c) obtained from the cDNA of the indicated cell subsets purified by FACS sorting from two HIV-1 patients (ID-96, ID-77). For each cell subsets RT-PCR for *GAPDH* was also performed as positive control of amplification (*bottom panels*). In both patients, a specific band of 500 bp was obtained from the cDNA extracted from the purified Treg and Tcm cell subsets. In ID_77 two different bands appeared in the agarose gel due to splice variants of the STAT5B gene. In PBMC the band is detectable only when 10-fold more cDNA was loaded in the RT-PCR. The band was not detected in all the other cell types, including Tscm, T effector memory (Tem), and others. **b** *Top panel*, agarose gel electrophoresis of the RT-PCR products obtained from the cDNA of the indicated cell subsets purified by magnetic cell isolation in four HIV-1 patients (ID-76, ID-56, ID-25, ID-63). RT-PCR for *GAPDH* was also performed as positive control of amplification (*bottom panel*). M: molecular size marker. **c**, **d** Histograms show the average of the relative levels of the copies of the HIV/*STAT5B* (**c**) and of the HIV/*BACH2* chimeric transcript (**d**) vs. *HPRT* copies measured by dd-PCR in Treg and Teff cells obtained from PBMC of six and three HIV-infected patients, respectively. **e**, **f** Histograms indicate the relative levels of the copies of *STAT5B* (**e**) and *BACH2* (**f**) vs. *HPRT* copies measured by dd-PCR in Treg and Teff cells obtained from PBMC of six and three HIV-infected patients, respectively. Significance was determined by Mann–Whitney tests (**p < 0.01), SEM is indicated by the *error bar*

silent nucleotide mutation was identified in a patient (ID_12), a significantly higher number of mutations ($N = 170$) were identified within the HIV-1 sequence ($p < 0.0001$ by Fisher exact test), indicating that the high rate of variability of the HIV-1 portion was not the result of sequencing errors (Supplementary Fig. 6 and Supplementary Table 6). In six patients multiple HIV-1 haplotypes were identified (from two up to four in a single patient), indicating that different HIV/STAT5B-expressing clones co-exist in patient's blood cells. Phylogenetic relationships among the distinct haplotypes identified within each patient were constructed using ClustalW2 program employing a neighbor-joining-based procedure (Fig. 4 and Supplementary Fig. 6). This analysis revealed that for patients ID_77 and ID_96 related viral sequences were present, indicating that different cell clones with HIV integrations targeting STAT5B were generated by evolutionary-related viral variants over time.

**BACH2 and STAT5B overexpression in primary T cells.** To address the impact of the expression of HIV/STAT5B and HIV/BACH2 chimeric transcripts in Treg cells, we overexpressed BACH2 or STAT5B in in vitro-induced Treg (iTreg)[24] and studied their immunophenotype and function. For this purpose, we designed bidirectional LVs allowing the coordinated expression of the Orange fluorescent protein (Or) with BACH2 (LV.BACH2) or STAT5B (LV.STAT5B) (Fig. 5a–c).

Naïve CD4$^+$ T cells, purified from PBMC of HDs, were activated, transduced with LV.BACH2, LV.STAT5B, or with a GFP-expressing LV as control, and then cultured in Treg polarizing condition (Fig. 5d, e)[24]. After 2 weeks of culture, no significant differences (by one-way ANOVA) in the frequency of CD25$^+$FOXP3$^+$ cells or in their ability to suppress the proliferation of activated allogeneic CD4$^+$CD25$^-$ Teff cells were found between STAT5B-expressing or BACH2-expressing cells and GFP-transduced or untransduced control cells (Fig. 5f, g). Thus, overexpression of STAT5B and BACH2 in naïve CD4$^+$ T cells neither affected their differentitation into iTreg cells nor their suppressive function. Interestingly, STAT5B-expressing iTreg cells were hyperproliferative upon IL-2 stimulation compared to GFP-transducedor untransduced control cells (Fig. 5h).

We next investigated the effect of BACH2 or STAT5B expression in primary Treg cells purified from HDs (Fig. 6a). The transduction of Treg cells with LV.BACH2 and LV.STAT5B

led to a 7.4-fold and 4.4-fold increase in the level of expression of BACH2 and STAT5B, respectively, when compared to GFP-transduced cells (Supplementary Fig. 7). After 10 days of culture, STAT5B-expressing and BACH2-expressing Treg cells showed a 2-fold increase in the total number of cells compared to controls (Mock) (Supplementary Fig. 7), but no significant differences in phenotype or function when compared to GFP-transduced or untransduced cells were observed (Fig. 6b and Supplementary Fig. 7).

To further characterize the selective advantage conferred by STAT5B and BACH2 expression, after the first 10 days of expansion, we performed a competitive proliferation experiment where primary Treg cells transduced with a GFP-expressing LV were mixed in a 1:1, 1:0.5 and 1:0.25 ratio with primary Treg cells transduced with BACH2-expressing or STAT5B-expressing LVs. The relative percentages of GFP$^+$ and Orange$^+$ cells were recorded over time after 10 and 20 days of culture. BACH2-expressing or STAT5B-expressing Treg cells significantly increased compared to GFP-expressing Treg cells, indicating that the expression of these transcription factors provides a significant selective advantage to Treg cells (Fig. 6c, d).

To assess if the selective advantage conferred by the over-expression of BACH2 or STAT5B was specific only for Treg cells, we tested the impact of their overexpression in primary CD4$^+$ naïve T cells purified from the blood of HDs. Primary CD4$^+$ naïve T cells were purified from six HDs, transduced with BACH2-expressing or STAT5B-expressing LVs at low MOI, to obtain a mixed population of transduced and non-transduced cells in the same well, and kept in culture in presence of low doses of IL-2 for 20 days. Orange-expressing cells at 10 and 20 days significantly increased compared to non-transduced cells, indicating that both transcription factors confer a proliferative advantage also to primary CD4$^+$ naïve T lymphocytes (Supplementary Fig. 8).

Finally, given that C-terminally truncated STAT5 isoforms, frequently detected in patients' PBMC[25], can act as repressors on HIV-1 transcription[26], we tested if the forced expression of STAT5B or BACH2 was able to decrease the transcriptional activity of an HIV-1 reporter genome in primary naïve CD4$^+$ T cells. From this analysis, we did not observe any significant variation in the transcriptional activity of our HIV-1 reporter genome (Supplementary Fig. 9).

## Discussion

Here, we show that in hematopoietic cells from a European cohort of HIV-1 patients under cART, BACH2 is significantly overtargeted by viral insertions oriented in the same direction of gene transcription. This finding is in agreement with three previous studies showing that BACH2 was highly targeted by viral insertions in PBMCs[9, 10] or CD4$^+$ T cells[8] from HIV-1 patients under cART as the result of insertional mutagenesis. Thus, our data reinforce the notion that BACH2 is a culprit of HIV-1 insertional mutagenesis and that it is reproducibly found overtargeted in independent cohorts of HIV-infected patients worldwide.

On the other hand, the integration study performed on our patient's cohort, together with the reanalysis of the integration data set published by Ikeda et al., candidates STAT5B as novel culprit of insertional mutagenesis in HIV patients under cART. Indeed, in our and in the Ikeda's integration site data sets, STAT5B was targeted at a significantly higher frequency than all other data sets and by viral insertions having almost in all the cases the same orientation of gene transcription. Maldarelli et al. proposed only BACH2 and MKL2 (targeted by 23 insertions, 22 of which in a single patient) as putative culprit of insertional mutagenesis[9]. However, MKL2 did not result to be significantly

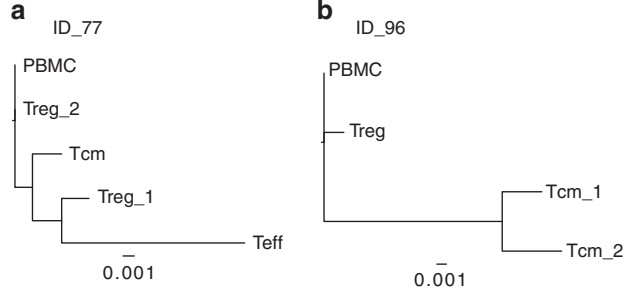

**Fig. 4** Phylogenies of amplified chimeric HIV/STAT5B bands. Maximum-likelihood trees constructed aligning the haplotypes identified analyzing the HIV-1 sequence from the U5 to the major SD signal among different hematopoietic cell subsets for patients ID_77 and ID_96. The hematopoietic cell subset from which the specific haplotype was identified is indicated at each branch tip of the tree. The progressive number indicates different haplotypes that have been identified within the same hematopoietic cell subset. Branch length varies according to the number of substitutions as a proportion of the length of the alignment (Distance value). *Scale bar* is reported below each tree

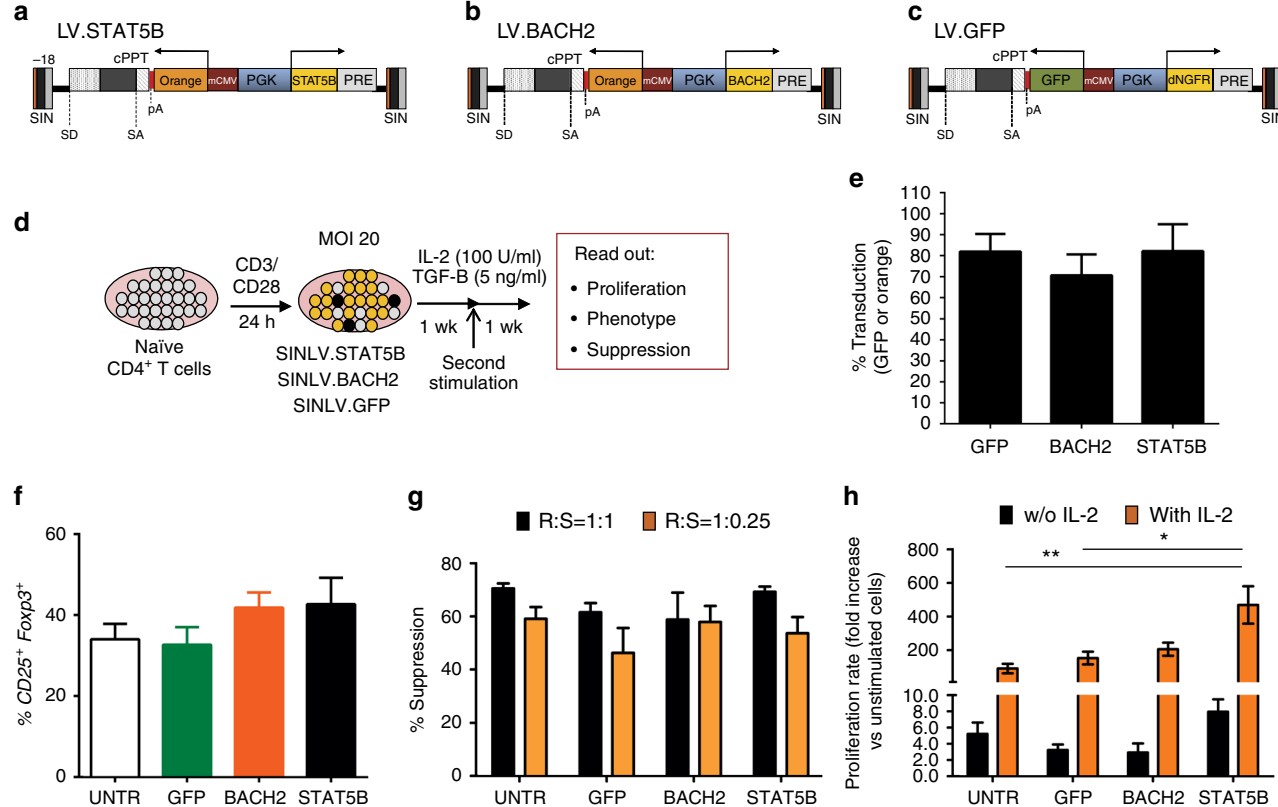

**Fig. 5** Effects of LV-mediated expression of STAT5B and BACH2 in in vitro-induced Treg cells. **a**–**c** Schematic structure of the LVs used to coordinately express the orange fluorescence protein (as marker transgene) and *STAT5B* (**a**) or *BACH2* (**b**) or the control vector-expressing GFP and dNGFR **c**. Transgene transcript is indicated by *arrows*. **d** Scheme of the experimental strategy. CD4+ naïve T cells from PBMC or cord blood mononuclear cells of HDs were activated by anti-CD3/CD28 beads for 24 h, transduced with the indicated LVs, and cultured in presence of IL-2 and TGF-β for 2 weeks; **e** Relative percentage of GFP+ or Orange+ cells (for bidirectional SIN LVs encoding for BACH2 or STAT5B) of naïve CD4+ T cells transduced with the indicated vector and analyzed 5 days after transduction, N = 5; **f** Percentage of CD25+FOXP3+ cells in CD4+ T cells transduced with the indicated vector and untransduced cells after 2 weeks of culture, N = 5; **g** Inhibition of proliferation (analyzed by flow cytometry for e-Fluor dilution and indicated as % suppression vs. responder alone) of anti-CD3/CD28 activated allogeneic Responder cells (R) cultured in presence of different doses (Ratio R:S were 1:1 and 1:0.25) of CD4+CD25+ suppressive cells (S) purified from cells transduced or not with the indicated LVs, N = 5; **h** Proliferation rate (measured by [3 H]-thymidine incorporation) of purified CD25+ cells from T cells transduced or not with the indicated vector and activated with anti-CD3 CD28 in the absence (*black bar*) or presence (*orange bar*) of IL-2. Fold increase vs. unstimulated control cells (N = 3) is shown, N = 5. Data are represented as Mean ± SEM. Significance was determined by one-way ANOVA with Bonferroni correction (*p < 0.05; **p < 0.01).

targeted by HIV-1 insertions neither in our data set nor in previous two studies[8, 10]. The reason why different studies identify different culprits of insertional mutagenesis is currently unclear.

Based on the previous knowledge linking viral/vector insertion patterns near or within cancer genes to molecular mechanisms known to interfere with their expression levels and/or with their mRNA structure[1, 20], we were able to "guess" the mechanism of insertional mutagenesis that is taking place when *STAT5B* and *BACH2* are targeted by HIV-1 insertions[3–5]. Indeed, the clustering of the viral insertions upstream the first coding exon and the orientation bias of the viral integrations in the same direction of gene transcription suggested that both these transcription factors could be activated by a promoter insertion mechanism where the expression of oncogenes is under the direct control of the promoter of the viral LTR, leading to their over-expression and/or to the production of aberrant proteins with enhanced oncogenic activity[1, 20]. Based on these premises, as an alternative approach to the much more cumbersome study of genomic integrations, we devised a nested RT-PCR strategy to specifically amplify the predicted chimeric HIV/*STAT5B* and HIV/*BACH2* transcripts. The RT-PCR approach allowed to readily detect the HIV/*STAT5B* and/or HIV/*BACH2* transcripts

in the mRNA obtained from PBMCs from 90 HIV patients. Interestingly, both chimeric transcripts were found in a relatively large proportion of patients (34%) and with similar frequencies in patients that experienced different cART regimens, thus suggesting that the treatment did not influence the rate of insertional mutagenesis events or the gene culprits implied in this phenomenon. Sequence analysis of the RT-PCR products showed that these mRNAs are predicted to encode for full-length wild-type STAT5B and BACH2 proteins. Unfortunately, only the identification of these chimeric mRNAs in purified cell clones harboring these specific integration events would formally demonstrate that they are produced as the result of insertional mutagenesis events. Nonetheless, sequence analysis of the RT-PCR products showed that all chimeric mRNAs identified from the different patients shared the same structure: starting from the HIV-1 LTR sequences, collinearly extended inside the viral genome up to the canonical SD consensus sequence and always correctly joined by splicing to the splice acceptor of the first coding exon of these two genes. The precision in which these chimeric transcripts follow the splicing rules indicates that the observed chimeric mRNAs are *bona fide* genuine aberrant transcripts generated by insertional mutagenesis rather than being PCR artifacts.

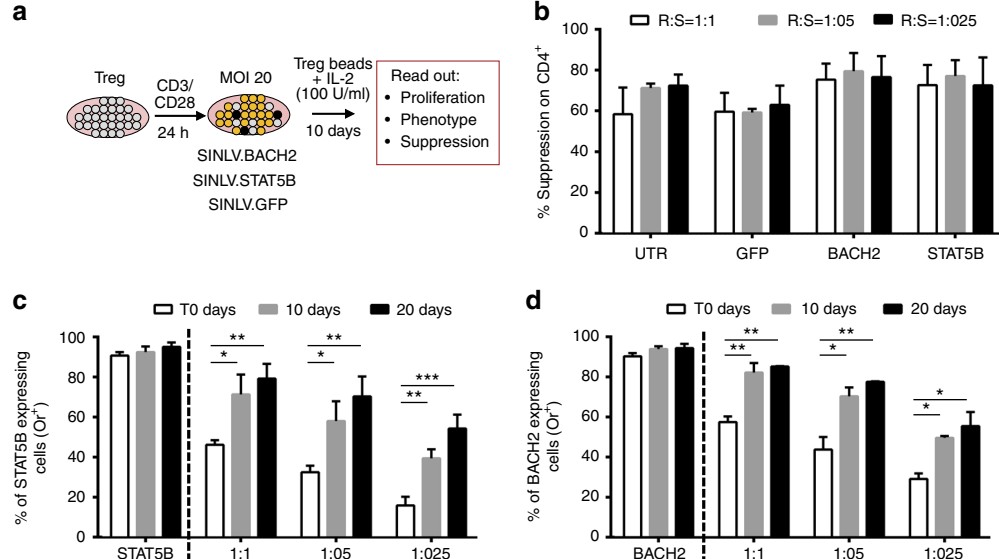

**Fig. 6** Effects of the forced expression of *STAT5B* and *BACH2* in Treg cells. **a** Experimental strategy. Treg cells were purified from PBMC of HDs, were activated by anti-CD3/CD28 Treg expander beads for 24 h, transduced with the indicated LVs, and cultured for 10 days in presence of Treg expander beads and IL-2 (100 U/ml). **b** Inhibition of proliferation (analyzed by flow cytometry for e-Fluor dilution and indicated as % Suppression vs. responder alone) of anti-CD3/CD28 activated allogeneic Responder (R) cells cultured in presence of different doses (Ratio R:S were 1:1, 1:05, and 1:0.25) of Suppressive cells (S). **c**, **d** Percentage of Orange⁺ Treg cells expressing STAT5B (**c**) or BACH2 (**d**) measured over time and cultured alone (indicated as STAT5B and BACH2) or in presence of Treg cells transduced with the control LV (**c**) (Ratio GFP-expressing and STAT5B-expressing cells is 1:1, 1:05, and 1:025, the same for GFP-expressing and BACH2-expressing cells). Significance was determined by performing two-way ANOVA with Bonferroni correction on Log ODD transformed % values (*$p < 0.05$; **$p < 0.01$). Data are represented as Mean ± SEM, and for all panels $N = 4$

Having addressed the feasibility of the RT-PCR to detect *bona fide* HIV/*STAT5B* and HIV/*BACH2* chimeric transcripts, we decided to assess their expression in different hematopoietic cell subsets. Strikingly, the RT-PCR and quantitative dd-PCR analysis revealed that these transcripts are enriched in Tcm and Treg cells but absent in CD8⁺ T cells or monocytes. Given that HIV/*STAT5B* and HIV/*BACH2* transcripts were found at two different time points, collected 6 years apart from the same patients (nine out of nine tested), the HIV-mediated activation of *STAT5B* or *BACH2* seems to provide a long-lasting selective advantage to Treg cells in HIV-infected patients. Moreover, the low level of expression of the HIV/*STAT5B* and HIV/*BACH2* transcripts compared to that of the relative host gene may suggest that cell clones harboring such activating HIV integrations are relatively rare when compared to the whole population tested. This speculation is in agreement with the data from Maldarelli et al. in which the relative abundance of expanded clones harboring insertional mutagenesis events in patients under cART was reported to be on average 2.3% of the total number of HIV-infected clones. Finally, by analyzing the sequences of the HIV/*STAT5B* aberrant transcripts in PBMCs and sorted cell populations, we found that mutations were mostly present in the viral sequence and rare in the *STAT5B* portion, indicating that these transcripts must be generated by independent cell clones infected by different viral haplotypes rather than being a consequence of point mutations acquired by the cell after integration. Importantly, in two patients the observed haplotypes were phylogenetically related, indicating that cell clones expressing these transcripts are continuously produced during viral evolution and persist overtime.

Notably, LV-mediated forced expression of the wild-type form of *STAT5B* or *BACH2* in primary Treg cells from HDs increased their proliferative capacity without affecting their phenotype and functions in vitro. Puzzlingly, forced expression of these transcription factors in primary CD4⁺ T cells appears to provide a

selective advantage in vitro. However, the reasons why the enrichment of these aberrant transcripts occurs only in Treg and Tcm lymphocyte subsets but not in other T cell lineages are currently unknown.

Although this study confirm the role of *BACH2* and candidates *STAT5B* as culprits of HIV-1-mediated insertional mutagenesis, shed light on the molecular mechanism used by HIV-1 insertions to alter their expression and identified the T cell subsets were these insertional mutagenesis events are enriched, several outstanding questions remain unanswered. It is unclear why no oncogenesis events were ever found associated to HIV-mediated insertional mutagenesis despite the high number of HIV-infected patients studied thus far, the huge number of integration events occurring during years of infection in each patient and the roles in cell survival and proliferation of *BACH2* and *STAT5B*. Indeed, these transcription factors control the function, homeostasis and development of several cell types of the immune system[27, 28], and the overexpression of their wild-type forms alone has never been associated to cancer. *STAT5B* can act as an oncogene only when specific activating mutations are present[29, 30], and *BACH2* is a tumor suppressor gene frequently found inactivated or silenced in human tumors[31–33], while been identified as CIS in murine insertional mutagenesis screenings[34]. Thus, the activation of these genes could possibly prompt a specific functional state that favors the long-term persistence of Treg cells without inducing their uncontrolled proliferation or cell transformation[35]. Moreover, beside *BACH2* and *STAT5B*, it is unclear why genes such as *LMO2*, *TAL1*, *CCND2*, *BMI1* and others, that are well-known players in spontaneously or vector-induced T cell leukemia in γ-retroviral vector-based HSC gene therapy patients[36–39], are not among the culprits of HIV-mediated insertional mutagenesis. We may speculate that several factors could be responsible for the specific outcome of HIV insertional mutagenesis. First, the occurrence and the type of tumor can be determined by the viral tropism and the cell-specific activity of the viral LTR, being

strongly modulated by the presence of cellular transcription factors that bind to the enhancer sequences contained within the LTR. For example, mice infected with replication-competent murine leukemia virus or mouse mammary tumor virus develop hematopoietic tumors or breast cancer, respeively, as the result of the different viral tropism and of the strong activity of the LTR in hematopoietic and epithelial progenitor/stem cells of these retroviruses[1]. Second, the genetic and/or disease background may also modulate the risk of cell transformation. Indeed, in γ-retroviral vector-based HSC clinical trials for the treatment of X-linked severe combined immunodeficiency[37, 38], Chronic granulomatous disease[39], and Wiskott Aldrich Syndrome[36], several patients developed leukemia as a consequence of vector-mediated insertional mutagenesis. However, the same vector type did not induce any transformation or clonal dominance events after >10 years of follow-up in a large cohort of HSC-transplanted patients enrolled in the gene therapy clinical trial for the treatment of adenosine deaminase SCID[40, 41]. Finally, in chronically infected HIV patients, transformed cells could suffer for cytotoxicity caused by the viral infection, thus preventing the neoplastic transformation of the infected cell. However, it has been shown that FOXP3 expression in Treg cells represses the HIV LTR activity by targeting the NF-kB pathway, thus making these cells less sensitive to the cytopathic effects of the viral protein compared to other cell lineages[42, 43]. Given these considerations we may speculate that Treg cell clones harboring HIV insertions driving the expression of BACH2 and STAT5B could escape from the lytic activity of HIV-specific CTLs, thanks to their suppressive ability. Therefore, the persistence of Treg cells harboring BACH2 and STAT5B activating insertions could be favored, thanks to the intrinsic resistance of this cell type to HIV cytotoxicity and to their natural ability to tame the immune system.

BACH2 and STAT5B are transcription factors that have an important role in the development and function of regulatory T cells. Indeed, STAT5B specifically propagates IL-2-mediated signal favoring the development and proliferation of Treg cells. Homozygous STAT5B-deficient patients showed reduced FOXP3 expression level and impaired Treg cell number and function[44], while transgenic mice expressing a constitutive active form of Stat5b were characterized by hyperactive and highly proliferating Treg cells[45, 46]. Regarding BACH2 and its role in the biology of Treg cells, it has been shown that it promotes the development and survival of Treg lineage, restrains aberrant differentiation of Treg cells, and protects against immune-mediated diseases[27, 47, 48]. Moreover, recent studies have highlighted important mechanistic roles for BACH2 in limiting terminal differentiation[47, 48], preventing T cell exhaustion[49], and potentially promoting the generation of long-lived, highly functional memory T cell population[49–51]. In support of these notions, Bach2[−/−] mice were characterized by a severely decreased number of functional FOXP3[+] Tregs that also displayed a terminally differentiated phenotype[47, 48]. Thus, considering the roles of BACH2 and STAT5B, the HIV-mediated overexpression of these genes could constitute part of the mechanism by which patients with HIV infection maintain a reservoir of infected cells in the Treg cell compartment. Treg cells are a relevant viral reservoir since they are long-lived, contain higher amounts of HIV DNA, and are able to produce more infectious viral particles upon activation compared to other T cell subsets[52]. We did not specifically address if cell clones harboring BACH2 or STAT5B activating viral insertions could produce infectious HIV particles. Although as recently reported the integrated provirus are frequently defective[53], rare expanded and persistent clones are able to produce infectious particles and contribute significantly to viral infection[54]. Based on these reports, we may speculate that rare long-lived cell clones with BACH2 or STAT5B activating viral insertions and harboring an infectious viral genome could exist and sustain viral infection during years. Another fascinating and non-mutually exclusive possibility is that the enriched Treg cell clones may actually be specific for viral antigens, thus their expansion could favor the formation of an HIV immune-tolerant milieu supporting viral persistence.

Overall, our molecular and functional studies provide novel compelling evidence that, as other retroviruses, HIV-1 takes advantage of insertional mutagenesis to favor its persistence in the host by infecting long-living and self-renewing cellular reservoir endowed with the ability to diminish the immune surveillance against infected cells. Our findings imply that HIV-1 does not only infect Treg cells but also favors their persistence over time, thereby "carving" its own reservoir. Given the role of STAT5B and BACH2 in promoting HIV-1 persistence, we propose that new targeted therapies aimed at interfering with their activity could provide the general means for the immunological re-sensitization toward HIV-1-infected cells and reduce long-lived cells carrying integrated, infectious proviruses with the ultimate goal of achieving a significant curtailment of latently infected cell reservoirs in cART-treated patients.

## Methods

**Study participants.** HIV-1 integration site analysis has been performed from total DNA extracted from PBMC of 54 patients that received a 3-drug cART regimen based on a virus protease inhibitor plus two nucleoside reverse-transcriptase inhibitors. Twenty-nine patients out of 54 (54%) received the same ART treatment plus IL-2 administration, as described in Tambussi[11].

For RNA studies, samples were obtained from 87 HIV-1-infected individuals enrolled in ART studies. Forty-eight subjects were NT who started cART based on their CD4[+] T cell counts, HIV-RNA levels in plasma (viremia) and the presence or absence of genotypic resistant mutations to antiretroviral agents, enrolled in previously published studies (MAIN study, EUDRACT number: 2008-007004-29; VEMAN study, EUDRACT number 2008-006287-11). Under cART, all patients achieved undetectable viremia levels (<50 copies of RNA/mL). Thirty-nine subjects were adult multi-experienced cART-treated HIV-1-infected patients who showed resistance or previous failures to all antiretroviral classes (NRTIs, NNRTIs, and PIs) and were enrolled into the MK0518-023, A4001050, and TMC125-C214 expanded access programs. A new cART regimen was prescribed based on genotypic resistant mutations, phenotypic viral tropism result, and previous failures history. All patients received raltegravir; new regimen was first long-term efficacious one, with achievement and maintenance of undetectable viremia levels. Informed consent has been signed from all patients. MAIN, VEMAN, MK0518-023, A4001050, and TMC125-C214 studies have been approved by the San Raffaele Hospital ethical committee.

**Genomic analysis.** Genomic DNA was extracted from total PBMC or cultured cells using the Qiagen-midi or micro DNA-Kit. For LV-transduced cells, vector copies per genome were determined by quantitative (q)PCR, as previously described[14]. LAM-PCR was executed as previously described[55] and used to retrieve HIV-1/genomic DNA junctions starting from 1 μg of DNA extracted from total PBMC. LAM-PCR products were sequenced by 454-Roche or Illumina MiSeq platforms and the sequences mapped on the human genome (Hg19, February 2009) using a dedicated bioinformatics pipeline[12–14].

**Cell purification, sorting, and flow cytometry.** To isolate hematopoietic cell subsets from HIV-1-infected patients, PBMC were isolated by Ficoll-Hypaque gradient from 90 ml of blood. Monocyte (CD14[+] CD3[−]), CD8[+], T-effector (CD4[+] CD25[−]), and Treg (CD4[+] CD25[+] CD127[−]) cells were isolated by FACS sorting procedure after staining of PBMC with fluorescent anti-CD3 (cod. 345765, 1:100 dilution, BD Biosciences), anti-CD4 (PRJNA392654), anti-CD8 (cod. 558116, 1:100 dilution, BD Biosciences), anti-CD14 (cod. 345784, 1:100 dilution, BD Biosciences), anti-CD25 (cod. 340907, 1:100 dilution, BD Biosciences), and anti-CD127 (cod. 25 12 78 42, 1:100 dilution, Ebioscience). T stem cell memory, T central memory, T naive, and T effector memory cells were isolated by FACS sorting procedure starting from an enriched fraction of CD4[+] cells purified using lineage-specific kit (Miltenyi Biotech). Next, CD4[+] cells were labeled with anti-CD8 (cod. 555368, 1:100 dilution, BD Biosciences), anti-CD45RA (cod. 560362, 1:100 dilution, BD Biosciences), anti-CD4 (cod.317418, 1:50 dilution, Biolegend), anti-CD95 (cod. 305608, 1:100 dilution, Biolegend), and anti-CD62L (cod. 1A-449-T100, 1:100 dilution, Exbio) fluorescent mAbs. The different T cell subsets were identified by the following markers: Tscm, CD4[+]CD45RA[+]

CD62L$^+$CD95$^+$; T naïve, CD4$^+$CD45RA$^+$CD62L$^+$CD95$^-$; Tcm, CD4$^+$CD45RA$^-$CD62L$^+$; and Tem, CD4$^+$CD45RA$^-$CD62L$^-$.

Cell sorting was performed by MoFlo-XDP cell sorter (BeckmanCoulter) and resulted in >97% purity, as determined by repeating flow cytometric analysis of some populations of the sorted cells. For other HIV-1-infected patients and healthy subjects, monocytes, CD8, Teff, and Treg cells were purified from total PBMCs by immune-magnetic separation using lineage-specific kits (Miltenyi Biotec). The purity of each fraction has been evaluated by flow cytometric analysis.

**Vector production and T cell transduction.** The bidirectional SIN.LV.PGK.BACH2 and SIN.LV.PGK.STAT5B plasmid constructs were generated using Gateway Technology (for details refer to Supplementary Methods). All concentrated VSV.G-pseudotyped LV stocks were produced by transient transfection of 293 T cells (CRL-3216, ATCC), as previously reported[7].

For transduction, Naïve T lymphocytes were pre-activated for 24 h in X-VIVO (Lonza), 5% human serum (Euroclone), in presence of anti-CD3/CD28 activating beads (Dynabeads Human T cell activator CD3/CD28, Life Technologies). Treg and Teff cells were pre-activated for 24 h in X-VIVO (Lonza), 5% human serum (Euroclone), in presence of Human Treg Expander beads (Life Technologies). Cells were infected with the indicated LV at MOI 20.

**Cell polarization, proliferation, and suppression assays.** Teff and Treg cells were purified from total PBMC from the blood of HIV-infected patients or from HD by immuno-magnetic cell purification. Purified cells were expanded in culture for 10 days by using the Dynabeads Human Treg Expander beads (Life Technologies) and rIL-2 (100 U/ml, Novartis). After 10 days of culture, the expression of CD25 (cod. 562442, 1:50 dilution, BD Bioscience) and FOXP3 (cod. 320214, 1:30 dilution, Biolegend) proteins were evaluated by FACS analyses within the expanded Treg and Teff cells. Furthermore, cell pellets from the expanded Treg and Teff were collected for further molecular biology analyses. T cell polarization, suppression, and proliferation assay on Treg cells have been performed as previously reported[24, 56, 57]. Briefly, human naïve CD4$^+$ cells were cultured in the presence of IL-2 (Novartis, 40 U/ml for naïve T cells), or with IL-2 and TGF-B (100 U/ml IL-2, Novartis and 5 ng/ml TGF-B, R&D Systems, Minneapolis, MN) to promote Treg polarization (iTreg condition). After 2 weeks of culture, cell phenotype was evaluated. The suppressive activity of ex vivo expanded Treg and iTreg cells were tested in an allogeneic setting. Indeed, allogeneic CD4$^+$ CD25$^-$ T cells or total PBMC were purified from normal donor, stained by e-Fluor (Invitrogen), activated with plate-bound anti-CD3 (10 μg/ml; OKT3, 130-093-387, Miltenyi) and soluble anti-CD28 mAbs (1 μg/ml, cod. 555725, BD Pharmingen) and cultured in presence or absence of different concentrations of iTreg and Treg. After 4–5 days of activation, proliferation was assessed by dilution of e-Fluor staining by flow cytometry. The proliferation of iTreg was evaluated on CD25$^+$ cells purified by magnetic immune-selection from transduced cells after 2 weeks of polarizing culture cells and activated, as described above, in presence or absence of IL2 (100 U/ml). After 72 h of culture, cells were pulsed for 16 h with [3 H] thymidine (1 mCi/well) (Amersham Biosciences).

**Gene expression analysis.** Total RNA was extracted from total PBMC or sorted cells with RNeasy purification kits (Qiagen) and reverse-transcribed with High Capacity cDNA Reverse Transcription Kit (Invitrogen). cDNA was used as template for qualitative-PCR.

Amplification of the chimerical HIV/STAT5B and HIV/BACH2 transcripts was obtained by two sequential nested PCR reactions using oligonucleotides complementary to the U3-HIV LTR and STAT5B exon4 or BACH2 exon6 and 7, see Supplementary Methods for sequence details.

Nested PCR was performed using 300 ng of cDNA isolated from 3 to $5 \times 10^6$ total PBMC using 300 ng of cDNA isolated from 3 to $5 \times 10^6$ total PBMC. The amplification was executed in a total volume of 25 μl and using GO Taq polymerase (Promega). The second nested PCR amplification was performed using 8 μl of the first PCR reaction. When PCR was performed on cDNA obtained from sorted cells, the first PCR reaction was executed using from 1 to 4 ng of starting material. Template-free controls and HIV negative samples were always used in our PCR testing and the presence of the specific chimeric bands in a sample was confirmed by repeating the PCR on newly retro-transcribed RNA from the same collection or on cDNA obtained from PBMC collected at a different time point. Furthermore, each amplified band was cloned and Sanger sequenced to confirm its viral/cellular chimeric nature (Supplementary Fig. 1). Finally, laboratory contaminations were excluded by looking for the presence of different point mutations among the different patients in the HIV portion of the chimeric sequence. Conversely, and as expected, the cellular genomic portion of BACH2 and STAT5B was conserved among the patient data set.

The expression levels of the HIV/STAT5B and HIV/BACH2 chimeric transcripts were measured by dd-PCR using the QuantaLife dd-PCR system. Custom-made dd-PCR assays were designed for the detection of the chimeric HIV/STAT5B and HIV/BACH2 transcripts (see Supplementary Methods for sequence details). TaqMan® Gene Expression Assays (Applied Biosystem) were used to assess gene expression of STAT5B (Hs00273500_m1, exon boundary 16-17), BACH2 (Hs00222364_m1, exon boundary 8–9), and HPRT

(Hs99999909_m1, exon boundary 6–7). Ten nanograms of cDNA were used for PCR amplification performed in duplicate and in a final volume of 20 μl. Approximately up to 20,000 monodispersed droplets for each sample were prepared using the QuantaLife droplet generator. Plates were quantified in a QuantaLife droplet reader, and the concentrations of the targets in the samples were determined using QuantaSoft software. The copies of the chimeric transcripts and of the host genes were normalized to HPRT copies. Significance was calculated using paired $t$-tests. See Supplementary Methods for further details.

On selected samples, Illumina barcodes were attached to the amplified products using TruSeq Nano DNA LT Sample Prep Kit (Illumina), and the different libraries were pooled and sequenced using an Illumina MySeq platform. Sequence reads were then aligned and a previously described approach[23] was used to identify the number and frequency of the different viral haplotypes contained in each library (see Supplementary Methods).

**Statistical analysis.** Statistical analyses were performed as indicated and using GraphPad Prism 5.0 software.

**Data availability.** The sequence of the RT-PCR products used to identify viral haplotypes and the raw sequencing data used to retrieve viral insertion sites are available from NCBI sequence archives under accession code PRJNA392654. The authors declare that all other data supporting the findings of this study are available within the article and its Supplementary Information files, or are available from the authors upon request.

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

## Acknowledgements

This work was supported by grants from the Italian Telethon Foundation to E.M. (TGT11D1, TGT16B01 and TGT16B03), Bill and Melinda Gates Foundation Grant Round 7 of Grand Challenges Explorations to E.M. (OPP1045909), and Giovani Ricercatori Grant 2013 from the Italian Ministry of Health to D.C. We are grateful to Luigi Naldini and Rosa Bacchetta for critical reading of the manuscript. We thank Andrea Galli, Grazia Locafaro, and Giulio Spinozzi for technical help and bioinformatics support. We acknowledge C. Villa and E. Canonico from Flow cytometry Resource and Advanced Cytometry Technical Applications Laboratory (FRACTAL) facility of OSR. We thank the patients for their participation in our studies.

## Author contributions

D.C. conceived the project, performed experiments, and wrote the manuscript. F.R.S.d.S. and L.P. provided technical support, discussed results, and reviewed the manuscript. S.G. designed the experiments on Treg cells, discussed results, and reviewed the manuscript. L.R., P.G. and E.B. performed experiments. A.C. performed the bioinformatics analyses of the HIV integration sites and I.M. and S.B. performed the phylogenetic analyses of the amplified chimeric sequence. E.V. and G.P. provided DNA material from HIV-1 patients for the integration studies and critically reviewed the manuscript, S.N. and G.T. provided all the clinical samples, coordinated the patients and critically reviewed the manuscript. E.M. conceived and supervised the project, designed the experiments, and wrote the manuscript.

## Additional information

**Competing interests:** The authors declare no competing financial interests.

