## [Peer Review File · Nature Communications]

Reviewers' comments:

Reviewer #1 (Remarks to the Author):

This study shows that in HIV patients under antiretroviral therapy there exists a hematopoietic cell population under biological selection, and that this selection is likely linked to provirus insertions into at least one of two genes: Bach2 and Stat5B. Authors show that the molecular mechanism contributing to this phenotype is aberrant splicing between a virus encoded splice donor site and a splice acceptor upstream of the first protein coding exons of these two genes. These chimeric transcripts are expected to encode full length Bach2 and Stat5B proteins. authors demonstrate a pronounced effect of overexpression of these two genes on the Treg compartment.

The study is well executed and highly informative with respect to clues for mechanisms of insertional mutagenesis by HIV. I have one technical and one other comment that relates to the presentation that authors should address in a revision.

1) Authors assume an overexpression of full length Bach2 and Stat5B proteins by the chimeric transcripts, but do not show this. In case patient sample material is still available, authors should nail protein overexpression by Bach2 and Stat5B antibodies in samples having the chimeric transcripts.

2) A discussion on Bach2 and Stat5B function with respect to downstream targets and pathways in light of selective persistence of Tregs would be highly appreciated and educational. In other words, how Bach2 or Stat5B overexpression is possibly linked to selective Treg proliferation and persistence?

Reviewer #2 (Remarks to the Author):

In this manuscript Dr. Montini and colleagues revealed that HIV-1 insertions targeting Bach2 and Stat5b are present in a significant number of patients under cART. The insertions lead to enhanced expression of full length Bach2 and Stat5b proteins, and are enriched in Tregs and central memory T cells. Ectopic expression of these proteins in Tregs can promote cell survival and proliferation. The authors concluded that these insertions could support a persistent viral reservoir in the Tregs of these patients. This is a well-written manuscript which this reviewer enjoyed reading. However, a major concern of this reviewer is the novelty of the findings. Although the discovery of HIV-1 insertion at the Stat5b locus is new, insertion at the Bach2 locus has been reported before (Maldarelli F. et al, Science 345,179-183 (2014)). Both Bach2 and Stat5b were known to promote Treg homeostasis (Kim EH et al, JI 192,985-995 (2014); Burchill MA et al, JI 178, 280-290 (2007)). It is also known that Tregs could serve as a persistent viral reservoir in patients under cART (Tran TA et al, PLOS One 3, e3305 (2008)). It is appreciated that the authors were able to link all these facts together along with their new findings on HIV-1 insertions at Stat5b, and made it into a coherent narrative. However, in this reviewer's opinion, the current manuscript does not meet the bar for publication in Nature Communication.

Reviewer #3 (Remarks to the Author):

Cesena et al. report the presence of integrations in the STAT5B and BACH2 genes, as well as production of chimeric HIV-Host RNA transcripts and propose a role for these chimeric transcripts in the development and maintenance of HIV reservoirs. Previously several investigators identified the presence of HIV integrations in BACH2 and STAT5B in cells from patients undergoing prolonged antiretroviral therapy and that integration played a role in persistence and expansion. Identifying a new mechanism for HIV persistence has made study of integration sites a very active area for research; nevertheless, proviral integration remains relatively poorly described, and new data are topical and critical to moving this field forward.

Here the investigators report the presence, in a number of patients, of a chimeric transcript including the HIV leader sequence spliced to the BACH2 and STAT5B genes. HIV proviruses have been noted in these genes previously, the current data reports that the integrated proviral sequence is expressed as a fusion transcript, these data are provocative and suggest persistence of HIV infected cells includes chimeric expression.

Patients used for study

Patient samples for integration analysis were from prior study of low dose intermittent subcutaneous IL-2 in individuals with detectable HIV virus loads. It seems that patients in this previous study were on stable antiretroviral therapy with for 18-21 months but had low level viremia, with mean viral RNA levels exceeding 1000 copies/ml plasma (ref 10) - is this correct? As these patients are not suppressed on therapy and may not be on effective cART, it is difficult to compare this group to previously reported analyses in which patients had been suppressed for prolonged periods. The presence of ongoing replication will confound the results, by introducing a large amount of unintegrated DNA into the analysis. The authors are therefore identifying integration sites as cells are continuing being infected; of note, this is the same circumstance as a previous group (Imamichi, et al, 2014) who also identified BACH2 integrants prior to HIV suppression on therapy. In addition, the authors have studied individuals who have received IL-2; while this patient group is quite interesting in itself, may be the cause of persistence or clonal expansion in these patients. These issues of patient selection and IL-2 use have not been sufficiently described in the ms. The authors perform a detailed comparison with previously published datasets from suppressed patients (lines 114-137, sup table 1 and 2), but comparing with their dataset derived from ongoing replication does not seem like a direct or appropriate comparison.

For the RNA analysis, line 395, a different cohort of patients, not IL-2 treated, were studied, including treatment experienced individuals who underwent new cART; as the authors describe, these patients achieved viral suppression, for varying periods. As these and other authors have described, it is essential to investigate patients who are suppressed on therapy because of the relevance of the studies to HIV persistence and cure.

The two sets of critical data for this study (DNA integration sites and chimeric RNA sequences) were obtained from different patients with different backgrounds, treatments, and outcomes, and the reader has to assume that the patients behave similarly. For instance, the authors find numerous BACH2 and STAT5B integration sites in incompletely

suppressed patients and they find BACH2-HIV fusion transcripts, in patients who have recently been suppressed on therapy; is it not clear that the integration sites in these patients are the same as the ones in the cohort used for RNA. The authors are making a fundamental, striking, and profound observation, studying DNA and RNA derived from the same PBMC is essential.

The authors make a strong case for insertional mutagenesis with the HIV provirus integrated in the same orientation as the host gene transcription, but in one case, STAT5B, the HIV provirus is integrated in the opposite orientation. Similarly, one of the BACH2 integrants seems to be downstream of the BACH2 ATG (supplementary figure 1A) The mechanisms for persistence or expansion of these proviruses is not described, but will not be the same as their main argument (splice based mechanism). Similarly, no data are reported on the proviral structures of the integrants in the respective BACH2 and STAT5B sites. The authors do not demonstrate that the integrants actually contain intact LTR and entire splice donor sequences. These retroviral features are assumed from the RNA sequencing data. Numerous chimeric transcripts have been reported in the past using RNA seq data, but confirmation of these putative transcripts is difficult and infrequently accomplished, and alternative hypotheses have been reported to explain chimeras. In the present case, the authors have detected the chimeras in numerous patients with identical splice site precision. These data are highly suggestive but more support for the source of these chimeric transcripts is necessary to make it definitive. The authors only present the results of a limited set of PCR primers, and more controls would be useful. Which LTR (5' or 3') is being used for this transcription, or is this chimeric RNA a read through? HIV primers in U3 (or nef) will help resolve the issue. Can the authors exclude the possibility that artefactual cDNA synthesis as described by Peng et al (J. Cancer, 2016) can be responsible for their observation. Do other HIV primers not in the LTR produce these transcripts? The authors report a large series of integration sites, in an analysis of many patients, but there do not seem to be many integration sites recovered per patient (Sup table 1). Why is the efficiency of detection so low? This assay has been used previously for identifying integration sites of integration in lentiviral therapeutic studies, where the vector has a uniform sequence. In the current case the HIV sequence may be highly variable, precluding efficient amplification. Is it possible that, in analysis of viremic patients, the excess unintegrated DNA is interfering with detection of integrants. Can the authors demonstrate whether any of these integrants have undergone clonal expansion? Also the authors report the exact same integrant in the gene LDLR at position 11231786 of chromosome 19 for two different patients (JID.19 and JID.83), suggesting cross patient contamination- is this an error?

Experiments of the forced, LV mediated expression of STAT5B and BACH2 (figure 5) are interesting and useful; the read out for these studies is GFP, however, and not the authentic BACH2 or STAT5B. As lentiviral integration can proceed with deletions, it is imperative that the authors demonstrate that they have introduced BACH2 and STAT5B and not just GFP
Minor:

- Line 37: "combined antiretroviral therapy" should read "combination antiretroviral therapy"

- Gene names BACH2 and STAT5B should be italicized throughout, some are not.

- The figure legend for figure 4 should explicitly describe panels G and H

- The authors correctly point out that most of the proviral integrants are noninfectious (line

374), and seem to speculate that an expanded integrant may encode infectious provirus and use refs 43 and 44 to buttress this argument. In fact, one of these references (44) reports that an expanded clone is infectious. This section of the ms requires clarification.

Answer to Reviewer 1:

This study shows that in HIV patients under antiretroviral therapy there exists a hematopoietic cell population under biological selection, and that this selection is likely linked to provirus insertions into at least one of two genes: Bach2 and Stat5B. Authors show that the molecular mechanism contributing to this phenotype is aberrant splicing between a virus encoded splice donor site and a splice acceptor upstream of the first protein coding exons of these two genes. These chimeric transcripts are expected to encode full length Bach2 and Stat5B proteins. Authors demonstrate a pronounced effect of overexpression of these two genes on the Treg compartment.

The study is well executed and highly informative with respect to clues for mechanisms of insertional mutagenesis by HIV. I have one technical and one other comment that relates to the presentation that authors should address in a revision.

We thank the Reviewer 1 for appreciating our study 'the study is well executed and highly informative with respect to clues for mechanisms of insertional mutagenesis by HIV'.

1) Authors assume an overexpression of full length Bach2 and Stat5B proteins by the chimeric transcripts, but do not show this. In case patient sample material is still available, authors should nail protein overexpression by Bach2 and Stat5B antibodies in samples having the chimeric transcripts.

*We agree that it would be very interesting to address protein overexpression of *BACH2* and *STAT5B* in samples having the chimeric transcripts. However, as shown by *Maldarelli et al.*, the relative abundance of the expanded clones harboring insertional mutagenesis events in patients under ART was on average 2.3%. Thus, the putative increase in protein levels caused by insertional mutagenesis on such small population will be negligible if tested on the entire population of PBMCs. Moreover, also our data suggest that the clones harboring activating HIV integrations are relatively rare since HIV/*BACH2* and HIV/*STAT5B* chimeric transcripts could be successfully amplified only when using nested RT-PCR or droplet digital PCR with 40,000 droplets (a number used to measure the expression of genes expressed at very low levels). On the other hand, single cell analyses could allow to evaluate the protein levels in cells harboring these insertions. However, since insertional mutagenesis events targeting *STAT5B* or *BACH2*, as predicted, should produce wild type proteins, it would be impossible to understand if single cells expressing high levels of any of these proteins are a consequence of insertional mutagenesis events or*

of physiological variations unrelated to viral insertions. Only the simultaneous isolation, from the same single cell, of DNA or RNA (to confirm the presence of the viral insertion within these genes) and proteins (to measure the protein levels) would solve this issue. Unfortunately the technology to study both DNA/RNA and proteins from the same single cell is very demanding and we are not sure how reliable would be the quantification in these settings. For these reasons we did not embark yet on this complex approach.

2) A discussion on Bach2 and Stat5B function with respect to downstream targets and pathways in light of selective persistence of Tregs would be highly appreciated and educational. In other words, how Bach2 or Stat5B overexpression is possibly linked to selective Treg proliferation and persistence?

We added a comment on the function in the revised form of the manuscript:

“BACH2 and STAT5B are transcription factors that have an important role in the development and function of regulatory T cells. STAT5B specifically propagates IL-2-mediated signal favoring the development and proliferation of Treg cells. Homozygous STAT5B-deficient patients showed reduced FOXP3 expression level and impaired Treg cell number and function, while transgenic mice expressing a constitutive active form of STAT5b were characterized by hyperactive and highly proliferating Treg cells. Regarding BACH2 and its role in Treg biology, it has been shown that it promotes the development and survival of Treg lineage, restrains aberrant differentiation of Treg cells, and protects against immune-mediated diseases. Recent evidences have highlighted important mechanistic roles for BACH2 in limiting terminal T cell differentiation, preventing T cell exhaustion and potentially promoting the generation of long-lived, highly functional memory T cell population. In support of these notions, Bach2^{-/-} mice were characterized by a severely decreased number of functional FOXP3⁺ Tregs. Thus, considering the roles of BACH2 and STAT5B, the HIV-mediated over-expression of these genes could be, at least in part, responsible for the persistence of a reservoir of HIV-1 infected cells in the Treg cell compartment”.

Answer to Reviewer 2:

In this manuscript Dr. Montini and colleagues revealed that HIV-1 insertions targeting Bach2 and Stat5b are present in a significant number of patients under cART. The insertions lead to enhanced expression of full length Bach2 and Stat5b proteins, and are enriched in Tregs and central memory T cells. Ectopic expression of these proteins in Tregs can promote cell survival and proliferation. The authors concluded that these insertions could support a persistent viral reservoir in the Tregs of these patients. ***This is a well-written manuscript which this reviewer enjoyed reading.*** However, a major concern of this reviewer is the novelty of the findings. Although the discovery of HIV-1 insertion at the Stat5b locus is new, insertion at the Bach2 locus has been reported before (Maldarelli F. et al, Science 345,179-183 (2014)). Both Bach2 and Stat5b were known to promote Treg homeostasis (Kim EH et al, JI 192,985-995 (2014); Burchill MA et al, JI 178, 280-290 (2007)). It is also known that Tregs could serve as a persistent viral reservoir in patients under cART (Tran TA et al, PLOS One 3, e3305 (2008)). It is appreciated that the authors were able to link all these facts together along with their new findings on HIV-1 insertions at Stat5b, and made it into a coherent narrative. However, in this reviewer's opinion, the current manuscript does not meet the bar for publication in Nature Communication.

We are pleased that the reviewer enjoyed this *well-written* manuscript. However we respectfully disagree with the considerations regarding the lack of novelty of our work and strongly disagree with the opinion that we just make '*coherent narrative*'.

The reasons why we think this manuscript is novel and relevant are the following:

- 1) As the reviewer itself pointed out, **the finding that *STAT5B* is a culprit of insertional mutagenesis is novel**. This finding is supported both by molecular data that we have produced and, of course, also by datamining of the previous literature that support our experimental data. The notion that *STAT5B* is well known to promote Treg homeostasis alone is not enough to make it a culprit of insertional mutagenesis of HIV-1. Actually the studies from *Maldarelli et al.*, and *Wagner et al.*, excluded *STAT5B* as a culprit of insertional mutagenesis. Therefore the involvement of *STAT5B* in this phenomenon is novel and not obvious.
- 2) The molecular mechanism involved in the deregulation of the already known target of insertional mutagenesis *BACH2* and the novel *STAT5B* was unknown so far. Although the orientation of these insertions with respect to these genes (same direction of gene transcription) and their position (almost all proviral insertions were upstream to the first coding exon) was suggestive of a mechanism of promoter insertion, no experimental evidences have been provided so far to sustain this "well educated guess". Based on these premises we developed a sensitive RT PCR strategy that allowed to analyze a relatively large European cohort of 87 HIV-1 infected individuals under two different cART regimens without going through the cumbersome integration site analysis approach. By this approach we found that HIV/*STAT5B* and HIV/*BACH2* chimeric transcripts were present in 34% of the tested individuals. These transcripts always contained HIV-1 sequences that started from the viral LTR, extended to the canonical HIV-1 donor splice site and correctly fused by splicing to the first protein-coding exon of these genes and predicted to encode for full-length unaltered proteins. These transcripts are unlikely be the product of a PCR artifact, since the HIV-1 sequences and cellular-gene exons of the chimeric transcripts were always correctly fused by the canonical rules of splicing and confirm the mechanism of promoter insertion. **Therefore, we provide experimental evidence that chimeric HIV/*STAT5B* and HIV/*BACH2* mRNAs are indeed present in the blood cells of these patients.** Moreover, we found that both chimeric transcripts were present with similar frequencies in patients that experienced different cART regimens, thus suggesting for the first time, that **the treatment did not influence the rate of insertional mutagenesis events or the gene culprits implied in this phenomenon**. Last but not the least, our estimation of the incidence of HIV-1-mediated insertional mutagenesis events was based on the study of a relatively large cohort of subjects (N= 54 and 87, screened at the DNA and RNA level respectively). Given that the previous estimations of the incidence of this phenomenon were based on a relatively small number of subjects (5 in *Maldarelli et al.*, and 3 in *Wagner et al.*), we think that our work provides a more reliable evaluation of how widespread is this phenomenon among HIV-1 infected individuals.

- 3) **It was not obvious that activation of these genes would confer a selective advantage to target cells.** Indeed, as we mentioned in the manuscript, it appears that *BACH2* inactivation as well as its activation could trigger cancer. On the other hand, *STAT5B* acts as an oncogene only when specific activating mutations are present. Thus also in this case the finding that the wild type form of *STAT5B* confers a selective advantage to cells with this type of insertions is novel and not obvious. Moreover, **we show that these aberrant transcripts are present specifically in T regulatory (Treg) and T central memory cells (Tcm)**, a finding that could be even considered as unexpected, since Tregs and Tcms are already long lived viral reservoirs. On the other hand, insertional mutagenesis in short lived T lymphocyte subsets or monocytes events, which has likely occurred, does not appear to provide a long lasting selective advantage to these cells.
- 4) **We experimentally demonstrated that both *BACH2* and *STAT5B* expression in human primary Treg cells significantly increased their proliferation and survival without compromising their immunosuppressive activity.** To our knowledge, these data are novel since the role of *BACH2* in promoting Treg homeostasis has been well established in mice, but less is known about its function in human cells. Regarding *STAT5B*, our data suggest that the wild type form of this gene, and not the mutated and hyper-activated one, was able to increase the proliferation rate of Treg cells.
- 5) In the improved version of our manuscript we took advantage of the high genetic diversity of the integrated HIV-1 genome to get insights on the number of different cellular clones expressing HIV/*STAT5B* chimeric transcripts in patients. Indeed, next generation sequencing of RT PCR products of HIV/*STAT5B* chimeric transcripts from 16 patients revealed that multiple mutations (haplotypes) were present only in the HIV-1 portion implying that the different transcript variants must have been generated by different viral variants that must have infected independent cell clones. By this analyses we identified that in 6 patients multiple HIV-1 haplotypes were found (from 2 to 4 variants for each patient), thus indicating that different HIV/*STAT5B* expressing clones co-exist in patient's blood cells. Moreover, phylogenetic analyses of the haplotypes identified within each patient revealed that for patients ID_77 and ID_96 related viral sequences were present suggesting that different clones with integrations targeting *STAT5B* were generated by different viral variants over time (See Figure 4, Suppl. Figure 6 and Supplementary Table 6). Phylogenetic analyses of HIV variants in infected patients are frequently used to study the evolution of HIV during infection but never to identify independent cell clones with specific insertional mutagenesis events. Thus we think that this approach is novel and tells that the generation of clones with insertional mutagenesis events occurs over time.

In conclusion, **we think that our work is original, relevant, supported by experimental data and not a mere 'coherent narrative' where we just link together different notions to write a nice story. We rather took advantage of the scientific notions available to formulate hypotheses and shed light on new biological aspects by experimental verification.**

Answer to Reviewer 3:

Cesana et al. report the presence of integrations in the STAT5B and BACH2 genes, as well as production of chimeric HIV-Host RNA transcripts and propose a role for these chimeric transcripts in the development and maintenance of HIV reservoirs. Previously several investigators identified the presence of HIV integrations in BACH2 and STAT5B in cells from patients undergoing prolonged antiretroviral therapy and that integration played a role in persistence and expansion. Identifying a new mechanism for HIV persistence has made study of integration sites a very active area for research; nevertheless, proviral integration remains relatively poorly described, and new data are topical and critical to moving this field forward.

Here the investigators report the presence, in a number of patients, of a chimeric transcript including the HIV leader sequence spliced to the BACH2 and STAT5B genes. HIV proviruses have been noted in these genes previously, the current data reports that the integrated proviral sequence is expressed as a fusion transcript, these data are provocative and suggest persistence of HIV infected cells includes chimeric expression.

1) Patients used for study

Patient samples for integration analysis were from prior study of low dose intermittent subcutaneous IL-2 in individuals with detectable HIV virus loads. It seems that patients in this previous study were on stable antiretroviral therapy with for 18-21 months but had low level viremia, with mean viral RNA levels exceeding 1000 copies/ml plasma (ref 10)- is this correct? As these patients are not suppressed on therapy and may not be on effective cART, it is difficult to compare this group to previously reported analyses in which patients had been suppressed for prolonged periods. The presence of ongoing replication will confound the results, by introducing a large amount of unintegrated DNA into the analysis. The authors are therefore identifying integration sites as cells are continuing being infected; of note, this is the same circumstance as a previous group (Imamichi, et al, 2014) who also identified BACH2 integrants prior to HIV suppression on therapy.

We agree with the Reviewer that the presence of ongoing viral replication, which results in high levels of unintegrated copies of viral DNA, could limit the efficiency of integration site retrieval. Moreover, the variations in the HIV-1 LTR sequence of the integrated provirus could prevent the binding of the oligonucleotides used in our study thus further reducing the efficiency of retrieval of integration sites. Both these factors could explain why, from 54 patients analyzed we obtained only 198 integration sites. However, besides the reduced efficiency, we did not observe any “qualitative” skewing in the integration profile. Indeed, the most targeted genes of our study, *STAT5B* and *BACH2*, were listed among the top 3 targeted genes in the studies from *Ikeda et al.*, *Maldarelli et al.*, *Wagner et al.*, that were carried in fully suppressed patients (See Suppl. Table 3). Moreover, the genomic distribution of HIV integration sites found in our dataset followed the well-known tendency of HIV-1 and replication-defective lentiviral vectors to integrate within gene bodies and gene dense regions and were distributed similarly to the integration datasets from *Maldarelli et al.*, *Wagner et al.*, according to Genome Hyperbrowser analysis (See Suppl. Figure 1 and Suppl. Table 2). Only in the dataset from *Ikeda et al.*, a

region on chromosome 6 spanning from nt 90,000 to 100,000,000 was significantly over targeted with respect to our dataset (FDR 10% adjusted p val= 1.6×10^{-5}). This difference was caused by a significant difference in the targeting frequency of the *BACH2* gene contained in that region. Thus, the strong similarity in the overall distribution of HIV insertion sites within the genome and among the list of the top targeted genes in our integration dataset and in all the other datasets analyzed, obtained from patients that were fully suppressed, suggest that the level of viral replication does not significantly impact on the results at the qualitative level.

To avoid the low efficiency issue and the cumbersome and time consuming procedures for integration site retrieval, we adopted the more efficient RT PCR strategy designed to amplify specific aberrant chimeric transcripts.

2) In addition, the authors have studied individuals who have received IL-2; while this patient group is quite interesting in itself, may be the cause of persistence or clonal expansion in these patients. These issues of patient selection and IL-2 use have not been sufficiently described in the ms. The authors perform a detailed comparison with previously published datasets from suppressed patients (lines 114-137, sup table 1 and 2), but comparing with their dataset derived from ongoing replication does not seem like a direct or appropriate comparison.

We think there is a misunderstanding, and we apologize for the lack of clarity. In our cohort, 29 patients out of 54 (54%) received IL-2 administration, while the other 25 patients (no IL-2 control group) did not. Although the 4 insertions (out of 121) targeting *STAT5B* were retrieved from the 29 patients that received the IL-2 treatment and none was retrieved from the integration dataset of the control group (without IL-2), this difference was not statistically significant (by Fisher exact test). Therefore, from this comparison, we cannot imply that IL-2 administration may be the cause of persistence or of clonal expansion in these patients. More convincingly, our RT PCR screening, performed on 87 fully suppressed patients that do not received IL-2 treatment, showed that 27 of them were positive for HIV/*STAT5B* chimeric transcripts. Similarly, HIV integrations targeting *STAT5B* have been identified also in the dataset from Ikeda *et al.*, obtained from fully suppressed patients that did not received any IL-2 treatment. Altogether, we do not have any direct evidence that the IL-2 treatment could favor the selection of clones harboring integration events activating *STAT5B*. In the main text and Material and Methods sections of the improved version of our manuscript we specified the type of treatment received by these patients and added an explanatory column in Suppl. Table 1.

3) For the RNA analysis, line 395, a different cohort of patients, not IL-2 treated, were studied, including treatment experienced individuals who underwent new cART; as the authors describe, these patients achieved viral suppression, for varying periods. As these and other authors have described, it is essential to investigate patients who are suppressed on therapy because of the relevance of the studies to HIV persistence and cure. The two sets of critical data for this study (DNA integration sites and chimeric RNA sequences) were obtained from different patients with different backgrounds, treatments, and outcomes, and the reader has to assume that the patients behave similarly. For instance, the authors find numerous *BACH2* and *STAT5B* integration sites in incompletely suppressed patients and they find *BACH2*-HIV fusion transcripts, in patients who have recently been suppressed on therapy; is it not clear

that the integration sites in these patients are the same as the ones in the cohort used for RNA. The authors are making a fundamental, striking, and profound observation, studying DNA and RNA derived from the same PBMC is essential.

The point is well taken, since and as pointed out in the main text of our manuscript, *'only the identification of these chimeric mRNAs from a single cell harboring these specific integration events would formally demonstrate that they are produced as the result of insertional mutagenesis events'*. Unfortunately, the simultaneous integration site retrieval and RT PCR analyses on single cell has not been developed so far and we do not know if the results would be reliable in qualitative and quantitative terms.

To mitigate this issue, in the new version of the manuscript we analyzed the RNA from PBMCs of 3 patients (the only available) that were previously analyzed for integration site analysis and that harbored insertions targeting *STAT5B* (JID.29 and JID.63) or *BACH2* (JID.87). The RT PCR approach used to amplify the HIV/*STAT5B* and HIV/*BACH2* chimeric transcripts provided the following results: HIV/*BACH2* chimeric transcript was found only in the patient JID.87 that harbored an insertion targeting *BACH2*, while HIV/*STAT5B* were instead found in all three patients (including JID.87 for which no insertions targeting *STAT5B* were retrieved). These results indicate that when the integrations are present, also the RT PCR product can be retrieved. Moreover, since in the patient JID.87, that harbored an insertion targeting *BACH2*, we found an aberrant transcript for *STAT5B*, this data suggest that the RT PCR is able to reveal insertional mutagenesis events that are instead missed by integration site retrieval.

4) The authors make a strong case for insertional mutagenesis with the HIV provirus integrated in the same orientation as the host gene transcription, but in one case, *STAT5B*, the HIV provirus is integrated in the opposite orientation. Similarly, one of the *BACH2* integrants seems to be downstream of the *BACH2* ATG (supplementary figure 1A). The mechanisms for persistence or expansion of these proviruses is not described, but will not be the same as their main argument (splice based mechanism). Similarly, no data are reported on the proviral structures of the integrants in the respective *BACH2* and *STAT5B* sites. The authors do not demonstrate that the integrants actually contain intact LTR and entire splice donor sequences. These retroviral features are assumed from the RNA sequencing data. Numerous chimeric transcripts have been reported in the past using RNA seq data, but confirmation of these putative transcripts is difficult and infrequently accomplished, and alternative hypotheses have been reported to explain chimeras. In the present case, the authors have detected the chimeras in numerous patients with identical splice site precision. These data are highly suggestive but more support for the source of these chimeric transcripts is necessary to make it definitive. The authors only present the results of a limited set of PCR primers, and more controls would be useful. Which LTR (5' or 3') is being used for this transcription, or is this chimeric RNA a read through? HIV primers in U3 (or nef) will help resolve the issue. Can the authors exclude the possibility that artefactual cDNA synthesis as described by Peng et al (J. Cancer, 2016) can be responsible for their observation. Do other HIV primers not in the LTR produce these transcripts?

The Reviewer is right since not all the viral insertions targeting these genes are suggestive or would necessarily result in gene activation by promoter insertion. This is expected since the insertions

retrieved by LAM PCR would not only retrieve insertions that conferred a selective advantage to the target cells but also those that are neutral (bystander insertions). Thus, to propose the presence of a promoter insertion mechanism of gene activation is enough to have a significant bias in the orientation of the viral insertions targeting these genes. The analysis of the integration dataset from our and from other studies (*Maldarelli et al.*, and *Ikeda et al.*), sustained this hypothesis. Regarding the insertion targeting *BACH2* after the first coding exon, we actually cannot exclude, nor confirm, that the putative chimeric transcript would still be produced and encode for an active protein (even if missing the first protein coding exon) conferring a selective advantage to the target cell. As shown in our previous studies, tumor prone *Cdkn2a*^{-/-} mice injected with a highly genotoxic lentiviral vector with active LTRs developed histiocytic sarcoma by insertional activation of *Braf* (*Cesana et al*, Mol Ther 2014). Almost all insertions were in the same orientation of *Braf* transcription (thus suggesting that *Braf* was activated by promoter insertion), however those insertions were not clustered upstream of the gene but rather in the middle (within intron 11 and 12) triggering the formation of a chimeric LV/*Braf* transcript encoding for a *Braf* form lacking the N-terminal regulatory domains. In all cases the first ATG codon available in the exons downstream the integration site was used as start codon. This truncated *Braf* protein, as reported in a transposon and lentiviral vector-based insertional mutagenesis studies, is constitutively active and oncogenic (*Collier et al.*, Nature 2005; *Ranzani et al.*, Nature Methods 2013; *Cesana et al.*, Molecular Therapy 2014).

Finally, the prediction that the mechanism of HIV-1 mediated activation of *STAT5B* and *BACH2* is achieved throughout the mechanism of promoter insertion was experimentally validated by RT PCR. The nested RT PCR strategy (using 2 different couples of oligonucleotides) was adopted to specifically amplify these chimeric transcripts. Each amplified PCR product was cloned and subjected to Sanger or NGS sequencing (in the new version of the manuscript). In all cases and for both chimeric transcripts, the transcript started from the viral 5' LTR sequence extended until the canonical viral splice donor signal and perfectly joined to the splice acceptor sites of *BACH2* and *STAT5B* exons. Only 34 % (N=30) of the tested patients resulted to be positive for at least of one of these transcripts. The presence of the specific chimeric transcripts in these patients was confirmed by repeating the PCR on newly retro-transcribed RNA from the same collection or on cDNA obtained from PBMC collected at a different time point. On the other hand negative patients remained negative when retested. Our template-free controls and HIV negative samples were always used in our PCR testing and always resulted negative. Considering the data described above we believe that our RT PCR is specific

5) The authors report a large series of integration sites, in an analysis of many patients, but there do not seem to be many integration sites recovered per patient (Sup table 1). Why is the efficiency of detection so low? This assay has been used previously for identifying integration sites of integration in lentiviral therapeutic studies, where the vector has a uniform sequence. In the current case the HIV sequence may be highly variable, precluding efficient amplification. Is it possible that, in analysis of viremic patients, the excess unintegrated DNA is interfering with detection of integrants.

As pointed out above, and as correctly indicated by the Reviewer, the efficiency of integration site retrieval observed in our cohort of 54 HIV infected patients it is low. It is possible that the presence of

unintegrated DNA and/or sequence variation within the HIV-1 LTR could reduce the efficiency of the amplification of the viral genomic junction. For this reasons our patient cohort was screened for the presence of chimeric HIV/*STAT5B*, /*BACH2* transcripts by RT-PCR, a more direct and sensitive method if compared to integration site retrieval by LAM PCR.

6) Can the authors demonstrate whether any of these integrants have undergone clonal expansion?

The integration site retrieval data can provide a rough estimation of the relative abundance of the clones harboring a specific insertion by evaluating the relative percent of sequence reads belonging to each specific insertion site with respect to the total number of reads identified for each patient. We did not perform this type of analysis because following widely accepted criteria in gene therapy to address if clonal expansion has occurred, the insertion should be retrieved in at least 3 consecutive time points and showed a consistent increase in the relative abundance of the traced integration site overtime. Since we had only one time point for this analysis and the number of insertions is low (sometimes only one integration in a single patient), we preferred to avoid to include this potentially misleading data.

Although we cannot provide a reliable estimation of the clonal abundance, we noticed that HIV/*BACH2* and HIV/*STAT5B* chimeric transcripts are amplified to detectable levels only when using nested PCR or droplet digital PCR with 40,000 droplets (a number used to measure the expression of genes expressed at low levels). This suggests that cell clones harboring activating HIV integrations are relatively rare and not very abundant when compared to the whole population tested. This speculation is in agreement with *Maldarelli et al.*, in which the relative abundance of the expanded clones harboring insertional mutagenesis events in patients under ART was estimated to be on average 2.3%.

7) Also the authors report the exact same integrant in the gene LDLR at position 11231786 of chromosome 19 for two different patients (JID.19 and JID.83), suggesting cross patient contamination- is this an error?

We thank the Reviewer for noticing this mistake. We found that the insertion belong to a single patient (the first analyzed). We eliminated the insertion from the other patient and made the new calculations based on the new number of integration sites (N=198) and corrected the main text and Supplementary Table1. Since only one insertion was eliminated the result of the new calculations are very similar and the conclusions identical to those reported in the previous version of our manuscript.

8) Experiments of the forced, LV mediated expression of *STAT5B* and *BACH2* (figure 5) are interesting and useful; the read out for these studies is GFP, however, and not the authentic *BACH2* or *STAT5B*. As lentiviral integration can proceed with deletions, it is imperative that the authors demonstrate that they have introduced *BACH2* and *STAT5B* and not just GFP

In the revised form of the manuscript we analyzed the level of expression of *BACH2* and *STAT5B* by ddPCR on cDNA obtained from Treg cells transduced with the different vectors (see Supplementary Figure 7C. Significant over-expression of *BACH2* ($p < 0.05$, N=7, 7.4 fold increase) and *STAT5B* ($p < 0.001$,

N=7, 4.4 fold increase) were observed in Treg cells transduced with LV.BACH2 and LV.STAT5B, when compared to GFP-transduced control cells.

Minor:

-Line 37: “combined antiretroviral therapy” should read “combination antiretroviral therapy”

We modified the text accordingly to Reviewer’s suggestion

-Gene names BACH2 and STAT5B should be italicized throughout, some are not.

Text format was changed accordingly

-The figure legend for figure 4 should explicitly describe panels G and H

We corrected this error

- The authors correctly point out that most of the proviral integrants are noninfectious (line 374), and seem to speculate that an expanded integrant may encode infectious provirus and use refs 43 and 44 to buttress this argument. In fact, one of these references (44) reports that an expanded clone is infectious. This section of the ms requires clarification.

We clarified this paragraph in the new version of the manuscript as follows:

‘Although as recently reported the integrated provirus are frequently defective (Cohn et al., Cell 2015), rare expanded and persistent clones are able to produce infectious particles and contribute significantly to viral infection (Simonetti F. R et al, PNAS 2016). Based on these reports we may speculate that rare long lived cell clones with BACH2 or STAT5B activating viral insertions and harboring an infectious viral genome could exist and sustain viral infection during years’.

REVIEWERS' COMMENTS:

Reviewer #1 (Remarks to the Author):

I appreciate authors' efforts and thorough explanations/additions in response to my and the other reviewers' critical observations. I am happy with the revised manuscript.

Reviewer #4 (Remarks to the Author):

Review of NCOMMS-16-18806A-Z "HIV mediated insertional activation..." by Montini and colleagues, addressing the question of whether the authors have adequately responded to comments of reviewer 3.

The authors follow up studies of others on expanded clones in HIV latency by showing chimeric transcripts between BACH2 and STAT5B and HIV, and further that the modified genes can drive proliferation of relevant cell types. In my opinion this is an important advance in understanding HIV latency.

Reviewer 3 pointed out several limitations of the study, for example the complex history of some of the patients, the use of several cohorts, and the lack of analysis of HIV-host splicing with further primer sets. However, there is no reason to doubt the basic measurements due to issues in the cohort used, and I thought the RNA mapping was convincing (known splice sites were involved in the cDNA chimeras mapped). I felt the authors responded to the comments of Reviewer 3 adequately.

A few minor points the authors may want to consider.

Use of "life cycle" invites an unresolvable debate on whether viruses are alive. I suggest "replication cycle".

Given that the authors are interested chimeric HIV-host messages, it might be useful to cite PMID:26377088, which reports RNAseq to study chimeric transcripts made by a low passage HIV isolate in T cells.

Answer to Reviewer 1:

I appreciate authors' efforts and thorough explanations/additions in response to my and the other reviewers' critical observations. I am happy with the revised manuscript.

We thank the Reviewer 1 for appreciating our study.

Answer to Reviewer 4:

Review of NCOMMS-16-18806A-Z “HIV mediated insertional activation...” by Montini and colleagues, addressing the question of whether the authors have adequately responded to comments of reviewer 3. The authors follow up studies of others on expanded clones in HIV latency by showing chimeric transcripts between BACH2 and STAT5B and HIV, and further that the modified genes can drive proliferation of relevant cell types. In my opinion this is an important advance in understanding HIV latency.

Reviewer 3 pointed out several limitations of the study, for example the complex history of some of the patients, the use of several cohorts, and the lack of analysis of HIV-host splicing with further primer sets. However, there is no reason to doubt the basic measurements due to issues in the cohort used, and I thought the RNA mapping was convincing (known splice sites were involved in the cDNA chimeras mapped). I felt the authors responded to the comments of Reviewer 3 adequately.

We thank Reviewer 4 for considering this work “an important advance in understanding HIV latency”.

A few minor points the authors may want to consider.

- 1) Use of “life cycle” invites an unresolvable debate on whether viruses are alive. I suggest “replication cycle”.

We modified the text accordingly to Reviewer’s suggestion

- 2) Given that the authors are interested chimeric HIV-host messages, it might be useful to cite PMID:26377088, which reports RNAseq to study chimeric transcripts made by a low passage HIV isolate in T cells.

We added the Reference in the Introduction and Discussion sections accordingly to Reviewer’s suggestion.